# Alternation makes the adversary weaker in two-player games

**Volkan Cevher**
LIONS, EPFL
volkan.cevher@epfl.ch

**Ashok Cutkosky**
Boston University
ashok@cutkosky.com

**Ali Kavis**
LIONS, EPFL
ali.kavis@epfl.ch

**Georgios Piliouras**
SUTD
georgios@sutd.edu.sg

**Stratis Skoulakis**
LIONS, EPFL
efstratios.skoulakis@epfl.ch

**Luca Viano**
LIONS, EPFL
luca.viano@epfl.ch

## Abstract

Motivated by alternating game-play in two-player games, we study an altenating variant of the *Online Linear Optimization* (OLO). In alternating OLO, a *learner* at each round $t \in [n]$ selects a vector $x^t$ and then an *adversary* selects a cost-vector $c^t \in [-1, 1]^n$. The learner then experiences cost $(c^t + c^{t-1})^\top x^t$ instead of $(c^t)^\top x^t$ as in standard OLO. We establish that under this small twist, the $\Omega(\sqrt{T})$ lower bound on the regret is no longer valid. More precisely, we present two online learning algorithms for alternating OLO that respectively admit $\mathcal{O}((\log n)^{4/3} T^{1/3})$ regret for the $n$-dimensional simplex and $\mathcal{O}(\rho \log T)$ regret for the ball of radius $\rho > 0$. Our results imply that in alternating game-play, an agent can always guarantee $\tilde{\mathcal{O}}((\log n)^{4/3} T^{1/3})$ regardless the strategies of the other agent while the regret bound improves to $\mathcal{O}(\log T)$ in case the agent admits only two actions.

## 1 Introduction

Game-dynamics study settings at which a set of selfish agents engaged in a repeated game *update* their strategies over time in their attempt to minimize their overall individual cost. In *simultaneous play* all agents simultaneously update their strategies, while in *alternating play* only one agent updates its strategy at each round while all the other agents stand still. Intuitively, each agent only updates its strategy *in response* to an observed change in another agent.

Alternating game-play captures interactions arising in various context such as animal behavior, social behavior, traffic networks etc. (see [33] for various interesting examples) and thus has received considerable attention from a game-theoretic point of view [11, 3, 33, 41, 40]. At the same time, *alternation* has been proven a valuable tool in tackling min-max problems arising in modern machine learning applications (e.g. training GANs, adversarial examples etc.) and thus has also been studied from an offline optimization perspective [37, 35, 21, 42, 9, 8, 10].

In the context of two-players, alternating game-play admits the following form: Alice (odd player) and Bob (even player) respectively update their strategies on odd and even rounds. Alice (resp. Bob) should select her strategy at an odd round so as to exploit Bob's strategy of the previous (even) round while at the same time protecting herself from Bob's response in the next (even) round. As a result, the following question arises:

***Q1:*** *How should Alice (resp. Bob) update her actions in the odd rounds so that, regardless of Bob's strategies, her overall cost (over the $T$ rounds of play) is minimized?*

## 1.1 Standard and Alternating Online Linear Minimization

Motivated by the above question and building on the recent line of research studying online learning settings with *restricted adversaries* [15, 25, 4, 5, 6, 34], we study an online linear optimization setting [43], called *alternating online linear optimization*. We use the term "*alternating*" to highlight the connection with alternating game-play that we subsequently present in Section 1.2.

In Algorithm 1 we jointly present both standard and alternating OLO so as to better illustrate the differences of the two settings.

---

**Algorithm 1** Standard and Alternating Online Linear Minimization

1: **Input**: A feasibility set $\mathcal{D} \subseteq \mathbb{R}^n$ and $c^0 \leftarrow (0, \ldots, 0)$.
2: **for each** round $t = 1, \ldots, T$ **do**
3:     The *learner* **selects** a vector $x^t \in \mathcal{D}$ based on $c^1, \ldots, c^{t-1} \in [-1, 1]^n$
4:     The *adversary* **learns** $x^t \in \mathcal{D}$ and **selects** a cost vector $c^t \in [-1, 1]^n$ (based on $x^1, \ldots, x^t$).
5:     The *learner* **learns** $c^t \in [-1, 1]^n$ and receives cost,

$$(c^t)^\top x^t \qquad \text{Standard OLM}$$

$$(c^t + c^{t-1})^\top x^t \quad \text{Alternating OLM}$$

6: **end for**

---

In both standard and alternating OLO, the adversary selects $c^t$ after the the learner's selection of $x^t$. The only difference between standard and alternating OLM is that in the first case the learner admits cost $(c^t)^\top x^t$ while in the second its cost is $(c^t + c^{t-1})^\top x^t$. An *online learning algorithm*[1] selects $x^t \in \mathcal{D}$ solely based on the previous cost-vector sequence $c^1, \ldots, c^{t-1} \in [-1, 1]^n$ with the goal minimizing the overall cost that is slightly different in standard and alternating OLO.

The quality of an online learning algorithm $\mathcal{A}$ in standard OLO is captured through the notion of *regret* [23], comparing $\mathcal{A}$'s overall cost with the overall cost of the *best fixed action*,

$$\mathcal{R}_{\mathcal{A}}(T) := \max_{c^1, \ldots, c^T} \left[ \sum_{t=1}^{T} (c^t)^\top x^t - \min_{x \in \mathcal{D}} \sum_{t=1}^{T} (c^t)^\top x \right]. \tag{1}$$

When $\mathcal{R}_{\mathcal{A}}(T) = o(T)$, the algorithm $\mathcal{A}$ is called *no-regret* since it ensured that regardless of the cost-vector sequence $c^1, \ldots, c^T$, the time-averaged overall cost of $\mathcal{A}$ approaches the time-averaged overall cost of the *best fixed action* with rate $o(T)/T \to 0$. Correspondingly, the quality of an online learning algorithm $\mathcal{A}$ in alternating OLO is captured through the notion of *alternating regret*,

$$\mathcal{R}_{\mathcal{A}}^{\text{alt}}(T) := \max_{c^1, \ldots, c^T} \left[ \sum_{t=1}^{T} (c^t + c^{t-1})^\top x^t - \min_{x \in \mathcal{D}} \sum_{t=1}^{T} (c^t + c^{t-1})^\top x \right]. \tag{2}$$

Over the years various no-regret algorithms have been proposed for different OLO settings[2] achieving $\mathcal{R}_{\mathcal{A}}(T) = \tilde{\mathcal{O}}\left(\sqrt{T}\right)$ regret [28, 20, 43]. The latter regret bounds are optimal since there is is a simple probabilistic construction establishing that any online learning algorithm $\mathcal{A}$ admits $\mathcal{R}_{\mathcal{A}}(T) = \Omega(\sqrt{T})$ even when $\mathcal{D}$ is the 2-dimensional simplex. This negative results comes from the fact that the adversary has access to the action $x^t$ of the algorithm and can appropriately select $c^t$ to maximize $\mathcal{A}$'s regret.

At a first sight, it may seem that the adversary can still enforce $\Omega\left(\sqrt{T}\right)$ alternating regret to any online learning algorithm $\mathcal{A}$ by appropriately selecting $c^t$ based on $x^t$ and possibly on $c^{t-1}$. Interestingly enough the construction establishing $\Omega(\sqrt{T})$ regret, fails in the case of alternating regret (see Section 2). As a result, the following question naturally arises,

---

[1] the notion of an online learning algorithm is exactly the same in standard and alternating OLO.
[2] the difference concerns the feasibility set $\mathcal{D}$.

***Q2***: *Are there online learning algorithm with $o\left(\sqrt{T}\right)$ alternating regret?*

Apart from its interest in the context of online learning, answering ***Q2*** implies a very sound answer to ***Q1***. In Section 1.2 we present the connection between Alternating OLO and Alternating Game-Play.

## 1.2 Alternating OLO and Alternating Game-Play

Alternating game-play in the context of two-player games can be described formally as follows: Let $(A, B)$ be a game played between Alice and Bob. The matrix $A \in [-1, 1]^{n \times m}$ represents Alice's costs, $A_{ij}$ is the cost of Alice if she selects action $i \in [n]$ and Bob selects action $j \in [m]$ (respectively $B \in [-1, 1]^{m \times n}$ for Bob). Initially Alice selects a mixed strategy $x^1 \in \Delta_n$. Then,

- At the even rounds $t = 2, 4, 6, \ldots, 2k$ : Bob plays a new mixed strategy $y^t \in \Delta_m$ and Alice plays $x^{t-1} \in \Delta_n$. Alice and Bob incur costs $(x^{t-1})^\top A y^t$ and $(y^t)^\top B x^{t-1}$ respectively.

- At the odd rounds $t = 3, 5, \ldots, 2k-1$ : Alice plays a new mixed strategy $x^t \in \Delta_n$ and Bob plays $y^{t-1} \in \Delta_m$. Alice and Bob incur costs $(x^t)^\top A y^{t-1}$ and $(y^{t-1})^\top B x^t$ respectively.

From the perspective of Alice (resp. Bob), the question is how to select her mixed strategies $x^1, x^3, \ldots, x^{2k-1} \in \Delta_n$ so as to minimize her overall cost

$$(x^1)^\top A y^2 + \sum_{k=1}^{T/2-1} (x^{2k+1})^\top A(y^{2k} + y^{2k+2}).$$

In Corollary 1.1 we establish that if Alice uses an online learning algorithm $\mathcal{A}$ then her overall regret (over the course of $T$ rounds of play) is at most $\mathcal{R}_{\mathcal{A}}^{\text{alt}}(T/2)$. As a result, in case ***Q2*** admits a positive answer, then Alice can guarantee at most $o(\sqrt{T})$ regret and improve over the $\tilde{\mathcal{O}}\left(\sqrt{T}\right)$ regret bound provided by standard no-regret algorithms [28, 20, 43, 23].

**Corollary 1.1.** *In case Alice (resp. Bob) uses an online learning algorithm $\mathcal{A}$ to update her strategies in the odd rounds, $x^{2k+1} := \mathcal{A}(Ay^2, Ay^4, \ldots, Ay^{2k})$ for $k = 1, \ldots, T/2-1$. Then no matter Bob's selected sequence $y^2, y^4, \ldots, y^T \in \Delta_m$,*

$$(x^1)^\top A y^2 + \sum_{k=1}^{T/2-1} (x^{2k+1})^\top A(y^{2k}+y^{2k+2}) - \min_{x \in \Delta_n} \left[ x^\top A y^2 + \sum_{k=1}^{T/2-1} x^\top A(y^{2k} + y^{2k+2}) \right] \le \mathcal{R}_{\mathcal{A}}^{\text{alt}}(T/2)$$

**Remark 1.2.** We remark that Corollary 1.1 refers to the standard notion of regret [23] and $\mathcal{R}_{\mathcal{A}}^{\text{alt}}(T/2)$ appears only as an upper bound. We additionally remark that if both Alice and Bob respectively use algorithms $\mathcal{A}$ and $\mathcal{B}$ in the context of alternating play, then the time-average strategy vector converges with rate $\mathcal{O}\left(\max(\mathcal{R}_{\mathcal{A}}(T), \mathcal{R}_{\mathcal{B}}(T))/T\right)$ to Nash Equilibrium in case of zero-sum games ($A = -B^\top$) and to Coarse Correlated Equilibrium for general two-player games [32]. Our objective is more general: we focus on optimizing the performance of a single player regardless of the actions of the other player.

## 1.3 Our Contribution and Techniques

In this work we answer ***Q2*** on the affirmative. More precisely we establish that,

- There exists an online learning algorithm (Algorithm 3) with alternating regret $\tilde{\mathcal{O}}\left((\log n)^{4/3} T^{1/3}\right)$ for $\mathcal{D} = \Delta_n$ ($n$-dimensional simplex).

- There exists an online learning algorithm (Algorithm 4) with alternating regret $\mathcal{O}\left(\rho \log T\right)$ for $\mathcal{D} = \mathbb{B}(c, \rho)$ (ball of radius $\rho$).

- There exists an online learning algorithm with alternating regret $\mathcal{O}\left(\log T\right)$ for $\mathcal{D} = \Delta_2$ (2-dimensional simplex), through a straight-forward reduction from $\mathcal{D} = \mathbb{B}(c, \rho)$.

Due to Corollary 1.1 our results provide a non-trivial answer to ***Q1*** and establish that Alice can substantially improve over the $\mathcal{O}(\sqrt{T})$ regret guarantees of standard no-regret algorithms.

**Corollary 1.3.** *In the context of alternating game play, Alice can always guarantee at most* $\tilde{\mathcal{O}}\left((\log n)^{4/3}T^{1/3}\right)$ *regret regardless the actions of Bob. Moreover in case Alice admits only* 2 *actions (n = 2), the regret bound improves to* $\mathcal{O}\left(\log T\right)$.

Bailey et al. [3] studied *alternating game-play in unconstrained two-player games* (the strategy space is $\mathbb{R}^n$ instead of $\Delta_n$). They established that if the $x$-player (resp. the $y$-player) uses *Online Gradient Descent* (OGD) with constant step-size $\gamma > 0$ ($x^{2k} := x^{2k-2} - \gamma A y^{2k-1}$) then it experiences at most $\mathcal{O}(1/\gamma)$ regret regardless the actions of the $y$-player. In the context of alternating OLM this result implies that OGD admits $\mathcal{O}(1/\gamma)$ alternating regret as long as *it always stays in the interior of* $\mathcal{D}$. However the latter cannot be guaranteed for bounded domains (simplex, ball). In fact there is a simple example for $\mathcal{D} = \Delta_2$ at which OGD with $\gamma$ step-size admits $\Omega(1/\gamma + \gamma T)$ alternating regret. More recently, [40] studied alternating game-play in zero-sum games ($B = -A^\top$). They established that if *both player* adopt Online Mirror Descent (OMD) the individual regret of each player is at most $\mathcal{O}(T^{1/3})$ and thus the time-averaged strategies converge to Nash Equilibrium with $\mathcal{O}(T^{-2/3})$ rate. The setting considered in this works differs because where the $y$-player can behave adversarially.

In order to achieve $\tilde{\mathcal{O}}\left((\log n)^{4/3}T^{1/3}\right)$ alternating regret in case $\mathcal{D} = \Delta_n$, we first propose an $\tilde{\mathcal{O}}(T^{1/3})$ algorithm for the special case of $\mathcal{D} = \Delta_2$. For this special case our proposed algorithm is an *optimistic-type* of *Follow the Regularized Leader* (FTRL) with *log-barrier regularization*. Using the latter as an algorithmic primitive, we derive the $\tilde{\mathcal{O}}\left((\log n)^{4/3}T^{1/3}\right)$ alternating regret algorithm for $\mathcal{D} = \Delta_n$, by upper bounding the overall alternating regret by the sum of *local alternating regret* of 2-actions decision points on a binary tree at which the leafs corresponds to the actual $n$ actions.

In order to achieve $\mathcal{O}(\rho \log T)$ alternating regret for $\mathcal{D} = \mathbb{B}(c, \rho)$ we follow a relatively different path. The major primitive of our algorithm is FTRL with adaptive step-size [16, 5]. The cornerstone of our approach is to establish that in case Adaptive FTRL admits more than $\mathcal{O}(\rho \log T)$ alternating regret, then *unormalized best-response* $(-c^{t-1})$ can compensate for the additional cost. By using a recent result on *Online Gradient Descent with Shrinking Domains* [5], we provide an algorithm interpolating between Adaptive FTRL and $-c^{t-1}$ that achieves $\mathcal{O}(\rho \log T)$ alternating regret.

### 1.4 Further Related Work

The question of going beyond $\mathcal{O}(\sqrt{T})$ regret in the context of *simultaneous game-play* has received a lot of attention. A recent line of work establishes that if both agents simultaneously use the *same no-regret algorithm* (in most cases Optimistic Hedge) to update their strategies, then the individual regret of each agent is $\tilde{\mathcal{O}}(1)$ [1, 14, 13, 2, 36, 26, 17].

Our work also relates with the more recent works in establishing improved regret bounds parametrized by the cost-vector sequence $c^1, \ldots, c^T$, sometimes also called "adaptive" regret bounds [16, 29, 38, 30, 12]. However these parametrized upper bounds focus on finding "easy" instances while still maintaining $\mathcal{O}(\sqrt{T})$ in the worst case. Alternating OLO can be considered as providing a slight "hint" to the learner that fundamentally changes the worst-case behavior, since its cost is $(c^t + c^{t-1})^\top x^t$ with the learner being aware of $c^{t-1}$ prior to selecting $x_t$. Improved regret bounds under different notions of hints have been established in [4, 5, 15, 34, 24, 39].

Finally our work also relates with the research line of no-regret learning in the context of *Extensive Form Games* [44, 18, 19] and *Stackelberg Games* [27, 22].

## 2 Preliminaries

We denote with $\Delta_n \subseteq \mathbb{R}^n$ the $n$-dimensional simplex, $\Delta_n := \{x \in \mathbb{R}^n : x_i \geq 0 \text{ and } \sum_{i=1}^n x_i = 1\}$. $\mathbb{B}(c, \rho)$ denotes the ball of radius $\rho > 0$ centered at $c \in \mathbb{R}^n$, $\mathbb{B}(c, \rho) := \{x \in \mathbb{R}^n : \|x - c\|_2 \leq \rho\}$. We also denote with $[x]_\mathcal{D} := \arg\min_{z \in \mathcal{D}} \|z - x\|^2$ the projection operator to set $\mathcal{D}$.

### 2.1 Standard and Alternating Online Linear Minimization

As depicted in Algorithm 1 the only difference between standard and Alternating OLM is the cost of the learner, $(c^t)^\top x^t$ (OLM) and $(c^t + c^{t-1})^\top x^t$ (Alternating OLM). Thus, the notion of an *online learning algorithm* is exactly the same in both settings.

**Definition 2.1.** An online learning algorithm $\mathcal{A}$, for an Online Linear Optimization setting with $\mathcal{D} \subseteq \mathbb{R}^n$, is a sequence of functions $\mathcal{A} := (\mathcal{A}_1, \ldots, \mathcal{A}_t, \ldots)$ where $\mathcal{A}_t : \underbrace{\mathbb{R}^d \times \ldots \times \mathbb{R}^d}_{t-1} \mapsto \mathcal{D}$.

As Definition 2.1 reveals, the notion of an online learning algorithm depends only on the feasibility set $\mathcal{D}$. As a result, an online learning algorithm $\mathcal{A}$ simultaneously admits both standard $\mathcal{R}_{\mathcal{A}}(T)$ and alternating regret $\mathcal{R}_{\mathcal{A}}^{\mathrm{alt}}(T)$ (see Equations 1 and 2 for the respective definitions). In Theorem 2.2, we present the well-known lower bound establishing that any online learning algorithm $\mathcal{A}$ admits $\mathcal{R}_{\mathcal{A}}(T) = \Omega(\sqrt{T})$ and explain why it fails in the case of alternating regret $\mathcal{R}_{\mathcal{A}}^{\mathrm{alt}}(T)$.

**Proposition 2.2.** *Any online learning algorithm $\mathcal{A}$ for $\mathcal{D} = \Delta_2$, admits regret $\mathcal{R}_{\mathcal{A}}(T) \geq \Omega\left(\sqrt{T}\right)$.*

*Proof.* Let $c^t$ be independently selected between $(-1, 1)$ and $(1, -1)$ with probability $1/2$. Since $c^t$ is independent of $(c^1, \ldots, c^{t-1})$ then $\sum_{t=1}^T \mathrm{E}\left[(c^t)^\top x^t\right] = 0$ where $x^t := \mathcal{A}_t(c^1, \ldots, c^{t-1})$. At the same time, $\mathrm{E}\left[-\min_{x \in \Delta_2} \sum_{t=1}^T (c^t)^\top x\right] \leq \mathcal{O}(\sqrt{T})$. As a result, $\mathcal{R}_{\mathcal{A}}(T) \geq \Omega(\sqrt{T})$. $\square$

We now explain why the above randomized construction does not apply for alternating regret $\mathcal{R}_{\mathcal{A}}^{\mathrm{alt}}(T)$. Let $\mathcal{A}$ be the *best-response algorithm*, $A_t(c^1, \ldots, c^{t-1}) := \operatorname{argmin}_{x \in \Delta_2}(c^{t-1})^\top x$. Since $c^t = (1, -1)$ or $c^t = (-1, 1)$ we get that $\min_{x \in \Delta_2}(c^{t-1})^\top x = -1$ while $\mathrm{E}\left[(c^t)^\top x^t\right] = 0$ since $x^t := \operatorname{argmin}_{x \in \Delta_2}(c^{t-1})^\top x$ and $c^t$ is independent of $c^{t-1}$. As a result,

$$\mathrm{E}\left[\sum_{t=1}^T (c^t + c^{t-1})^\top x^t - \sum_{t=1}^T \min_{x \in \Delta_2}(c^t + c^{t-1})^\top x\right] = -T + \Omega(\sqrt{T}).$$

The latter implies that there exists at least one online learning algorithm (*Best-Response*) that admits $\Theta(-T)$ alternating regret in the above randomized construction. However the latter is not very informative since there is a simple construction at which *Best-Response* admits linear alternating regret.

We conclude this section with the formal statement of our results. First, for the case that $\mathcal{D}$ is the simplex, we show $\tilde{O}(T^{1/3})$ alternating regret (Section 3):

**Theorem 2.3.** *Let $\mathcal{D}$ be the $n$-dimensional simplex, $\mathcal{D} = \Delta_n$. There exists an online learning algorithm $\mathcal{A}$ (Algorithm 3) such that for any cost-vector sequence $c^1, \ldots, c^T \in [-1, 1]^n$,*

$$\sum_{t=1}^T (c^{t-1} + c^t)^\top x^t - \min_{x^* \in \mathcal{D}} \sum_{t=1}^T (c^{t-1} + c^t)^\top x^* \leq \mathcal{O}\left(T^{1/3} \cdot \log^{4/3}(nT)\right) \text{ where } x^t = \mathcal{A}_t(c^1, \ldots, c^{t-1}).$$

Next, when $\mathcal{D}$ is a ball of radius $\rho$, we can improve to $\tilde{O}(1)$ alternating regret (Section 4):

**Theorem 2.4.** *Let $\mathcal{D}$ be a ball of radius $\rho$, $\mathcal{D} = \mathbb{B}(c, \rho)$. There exists an online learning algorithm $\mathcal{A}$ (Algorithm 4) such that for any cost-vector sequence $c^1, \ldots, c^T$ where $\|c^t\|_2 \leq 1$,*

$$\sum_{t=1}^T (c^{t-1} + c^t)^\top x^t - \min_{x^* \in \mathcal{D}} \sum_{t=1}^T (c^{t-1} + c^t)^\top x^* \leq \mathcal{O}(\rho \log T) \quad \text{where} \quad x^t = \mathcal{A}_t(c^1, \ldots, c^{t-1}).$$

**Remark 2.5.** Using Algorithm 4 we directly get an online learning algorithm with $\mathcal{O}(\log T)$ alternating regret for $\mathcal{D} = \Delta_2$.

## 2.2 Alternating Game-Play

A *two-player normal form game* $(A, B)$ is defined by the payoff matrix $A \in [-1, 1]^{n \times m}$ denoting the payoff of Alice and the matrix $B \in [-1, 1]^{m \times n}$ denoting the payoff of Bob. Once the Alice selects a mixed strategy $x \in \Delta_n$ (prob. distr. over $[n]$) and Bob selects a mixed strategy $y \in \Delta_m$ (prob. distr. over $[n]$). Then Alice suffers (expected) cost $x^\top A y$ and Bob $y^\top B x$.

In alternating game-play, Alice updates her mixed strategy in the even rounds while Bob updates in the odd rounds. As a result, a sequence of alternating play for $T = 2K$ rounds (resp. for $T = 2K + 1$) admits the form $(x^1, y^2), (x^3, y^2), \ldots, (x^{2k+1}, y^{2k}), (x^{2k+1}, y^{2k+2}), \ldots, (x^{2K-1}, y^{2K})$. Thus, the *regret* of Alice in the above sequence of play equals the difference between her overall cost and the cost of the *best-fixed action*,

$$\mathcal{R}_x(T) := \underbrace{(x^1)^\top A y^2 + \sum_{k=1}^{T/2-1} (x^{2k+1})^\top A(y^{2k} + y^{2k+2})}_{\text{Alice's cost}} - \underbrace{\min_{x \in \Delta_n}\left[x^\top A y^1 + \sum_{k=1}^{T/2-1} x^\top A(y^{2k} + y^{2k+2})\right]}_{\text{cost of Alice's best action}}$$

If Alice selects $x^{2k+1} := \mathcal{A}_k(Ay^2, Ay^4, \ldots, Ay^{2k-2}, Ay^{2k})$ for $k \in [K-1]$ and $x_1 = \mathcal{A}_1(\cdot)$ then by the definition of alternating regret in Equation 2, we get that

$$(x^1)^\top Ay^2 + \sum_{k=1}^{K-1} (x^{2k+1})^\top (Ay^{2k} + Ay^{2k+2}) - \min_{x \in \Delta_n} \left[ x^\top Ay^2 + \sum_{k=1}^{K-1} x^\top (Ay^{2k} + Ay^{2k+2}) \right] \leq \mathcal{R}_{\mathcal{A}}^{\mathrm{alt}}(K)$$

which establishes Corollary 1.1. The proof for $T = 2K+1$ is the same by considering $Ay^{2K+2} = 0$.

## 3 The Simplex case

Before presenting our algorithm for the $n$-dimensional simplex, we present Algorithm 2 that admits $\mathcal{O}(\log^{2/3} T \cdot T^{1/3})$ alternating regret for the 2-simplex and is the basis of our algorithm for $\Delta_n$.

**Definition 3.1** (Log-Barrier Regularization). Let the function $R : \Delta_2 \mapsto \mathbb{R}_{\geq 0}$ where $R(x) := -\log x_1 - \log x_2$.

---

**Algorithm 2** Online Learning Algorithm for 2D-Simplex

---

1: **Input:** $c^0 \leftarrow (0,0)$
2: **for** rounds $t = 1, \ldots, T$ **do**
3:     The learner **selects** $x^t := \min_{x \in \Delta_2} [2(c^{t-1})^\top x + \sum_{\tau=1}^{t-1} (c^\tau + c^{\tau-1})^\top x + R(x)/\gamma]$.
4:     The adversary **selects** cost vector $c^t \in [0,1]^n$
5:     The learner **suffers** cost $(c^t + c^{t-1})^\top x^t$
6: **end for**

---

In order to analyze Algorithm 2 we will compare its performance with the performance of the *Be the Regularized Leader algorithm* with *log-barrier regularization* that is ensured to achieve $\mathcal{O}(\log T/\gamma)$ alternating regret [23]. The latter is formally stated and established in Lemma 3.2.

**Lemma 3.2.** Let $y^1, \ldots, y^T \in \Delta_2$ where $y^t := \min_{x \in \Delta_2} \left[ (c^t + c^{t-1})^\top x + \sum_{s=1}^{t-1} (c^s + c^{s-1})^\top x + R(x)/\gamma \right]$. Then, $\sum_{t=1}^T (c^t + c^{t-1})^\top y^t - \min_{i \in [n]} \sum_{t=1}^T (c_i^t + c_i^{t-1}) \leq 2\log T/\gamma + 2$.

In Lemma 3.3 we provide a closed formula capturing the difference between the output $x^t \in \Delta_2$ of Algorithm 2 and the output $y^t \in \Delta_2$ of *Be the Regularized Leader algorithm* defined in Lemma 3.2.

**Lemma 3.3.** Let $x^t = (x_1^t, x_2^t) \in \Delta_2$ as in Algorithm 2 and $y^t = (y_1^t, y_2^t) \in \Delta_2$ as in Lemma 3.2. Then,
$$x_1^t - y_1^t = \gamma A^{-1}(x_1^t, y_1^t) \cdot \left( (c_1^t - c_2^t) - (c_1^{t-1} - c_2^{t-1}) \right)$$
with $A(x_1, y_1) := (x_1 y_1)^{-1} + (1-x_1)^{-1}(1-y_1)^{-1}$ and $|A^{-1}(x_1^t, y_1^t) - A^{-1}(x_1^{t+1}, y_1^{t+1})| \leq \mathcal{O}(\gamma)$.

Up next we use Lemma 3.2 and Lemma 3.3 to establish that Algorithm 2 admits $\mathcal{O}(\log^{2/3} T \cdot T^{1/3})$ alternating regret.

**Theorem 3.4.** Let $x^1, \ldots, x^T \in \Delta_2$ the sequence produced by Algorithm 2 for the cost sequence $c^1, \ldots, c^T \in [-1,1]^2$ with $\gamma = \mathcal{O}\left( \log^{1/3} T \cdot T^{-1/3} \right)$ then $\mathcal{R}^{\mathrm{alt}}(T) = \mathcal{O}\left( \log^{2/3} T \cdot T^{1/3} \right)$.

*Proof.* By Lemma 3.2 then $\sum_{t \in [T]} (c^t + c^{t-1})^\top x^t - \min_{i \in [n]} \sum_{t \in [T]} (c_i^t + c_i^{t-1}) \leq \mathcal{O}(\log T/\gamma) + \sum_{t \in [T]} (c^t + c^{t-1})^\top (x^t - y^t)$ where $y^t \in \Delta_2$ as in Lemma 3.2. Using Lemma 3.3 we get that

$$\sum_{t=1}^T (c^t + c^{t-1})^\top (x^t - y^t) = \sum_{t=1}^T \left( (c_1^t - c_2^t) + (c_1^{t-1} - c_2^{t-1}) \right) (x_1^t - y_1^t)$$

$$= \gamma \sum_{t=1}^T \left( (c_1^t - c_2^t) + (c_1^{t-1} - c_2^{t-1}) \right) A^{-1}(x_1^t, y_1^t) \cdot \left( (c_1^t - c_2^t) - (c_1^{t-1} - c_2^{t-1}) \right)$$

$$= \gamma \sum_{t=1}^T A^{-1}(x_1^t, y_1^t) \left( (c_1^t - c_2^t)^2 - (c_1^{t-1} - c_2^{t-1})^2 \right)$$

$$= \gamma \sum_{t=1}^T (c_1^t - c_2^t)^2 \cdot \left( A^{-1}(x_1^t, y_1^t) - A^{-1}(x_1^{t+1}, y_1^{t+1}) \right) \leq \mathcal{O}(\gamma^2 T)$$

Hence $\mathcal{R}_{alt}(T) \leq \mathcal{O}\left( \log T/\gamma + \gamma^2 T \right) \leq \mathcal{O}\left( \log^{2/3} T \cdot T^{1/3} \right)$ for $\gamma := \mathcal{O}\left( \log^{1/3} T/T^{1/3} \right)$. $\square$

## 3.1 The $n$-Dimensional Simplex

In this section we extend Algorithm 2 to the case of the $n$-dimensional simplex. Our extension is motivated and builds upon the CFR algorithm develloped in the context of EFGs [44].

Without loss of generality we assume that $n = 2^H$. We consider a complete binary tree $T(V, E)$ of height $H = \log n$ where the *leaves* $L \subseteq V$ corresponds to the $n$ *actions*, $|L| = n$. Each node $s \in V/L$ admits exactly two children with $\ell(s), r(s)$ respectively denoting the left and right child. Moreover, Level(h) $\subseteq V$ denotes the nodes lying at depth $h$ from the root ( Level(1) = {root} and Level($\log n$) = $L$). Up next we present the notion of *policy* on the nodes of $T(V, E)$.

**Definition 3.5.** • A policy over the nodes $\pi : V/L \mapsto \Delta_2$ encodes the probability of selecting the left/right child at node $s \in V$. Specifically $\pi(s) = (\pi(\ell(s)|s), \pi(r(s)|s))$ where $\pi(\ell(s)|s) + \pi(r(s)|s) = 1$ and $\pi(\ell(s)|s)$ is the probability of selecting $\ell(s)$ (resp. for $r(s)$).

• $\mathrm{Pr}(s, i, \pi)$ denotes the probability of reaching leaf $i \in L$ starting from node $s \in V/L$ and following $\pi(\cdot)$ at each step.

• $x^\pi \in \Delta_n$ denotes the probability distribution over the *leaves/actions* induced by $\pi(\cdot)$. Formally, we have $x_i^\pi := \mathrm{Pr}(\text{root}, i, \pi)$ for each leaf $i \in L$.

**Definition 3.6.** Given a cost vector $c \in [-1, 1]^n$ for the *leaves/actions*, the *virtual cost* of a node $s \in V$ under policy $\pi(\cdot)$, denoted as $Q(s, \pi, c)$, equals

$$Q(s, \pi, c) := \left\{ \begin{array}{ll} c_s & s \in L \\ \sum_{i \in L} \mathrm{Pr}(s, i, \pi) \cdot c_i & s \notin L \end{array} \right.$$

The *virtual cost vector* of $s \in V$ under $\pi(\cdot)$ is defined as $q(s, \pi, c) := (Q(\ell(s), \pi, c), Q(r(s), \pi, c))$.

We remark that $Q(s, \pi, c)$ is the *expected cost* of the random walk starting from $s \in V$ and following policy $\pi(\cdot)$ until a leaf $i \in L$ is reached in which case cost $c_i$ is occurred.

Our online learning algorithm for the $n$-dimensional simplex is illustrated in Algorithm 3.

---

**Algorithm 3** An Online Learning Algorithm for the $n$-Dimensional Simplex

1: **Input:** A sequence of cost vectors $c^1, \ldots, c^T \in [-1, 1]^n$
2: The learner constructs a complete binary tree $T(V, E)$ with $L = \mathcal{A}$.
3: **for** each round $t = 1, \ldots, T$ **do**
4:     **for** each $h = \log n$ to 1 **do**
5:         **for** every node $s \in \text{Level}(h)$ **do**
6:             The learner computes $q\left(s, \pi^t, c^{t-1}\right) := \left(Q\left(\ell(s), \pi^t, c^{t-1}\right), Q\left(r(s), \pi^t, c^{t-1}\right)\right)$ and sets

$$\pi^t(s) := \arg\min_{x \in \Delta_2} \left[ 2q(s, \pi^t, c^{t-1})^\top x + \sum_{\tau=1}^{t-1} \left( q(s, \pi^\tau, c^{\tau-1}) + q(s, \pi^\tau, c^\tau) \right)^\top x + R(x)/\gamma \right]$$

7:         **end for**
8:     **end for**
9:     The learner *selects* $x^t := x^{\pi^t} \in \Delta_n$ (induced by policy $\pi_t$, Definition 3.5).
10:     The adversary *selects* cost vector $c^t \in [0, 1]^n$
11:     The learner *suffers* cost $(c^t + c^{t-1})^\top y^t$
12: **end for**

---

We remark that at each round $t$, the learner computes a policy $\pi^t(\cdot)$ as an intermediate step (Step 6) that then uses to select the probability distribution $x^t := x^{\pi^t} \in \Delta_n$ (Step 9). Notice that the computation of policy $\pi^t(\cdot)$ is performed in Steps (4)-(8). Since nodes are processed in decreasing order (with respect to their level), during Step 6 $\pi^t(\cdot)$ has already been determined for nodes $\ell(s), r(s)$ and thus $Q\left(\ell(s), \pi^t, c^{t-1}\right), Q\left(r(s), \pi^t, c^{t-1}\right)$ are well-defined.

Up next we present the main steps for establishing Theorem 2.3. A key notion in the analysis of Algorithm 3 is that of *local alternating regret* of a node $s \in V$ presented in Definition 3.7. As established in Lemma 3.8 the overall alternating regret of Algorithm 3 can be upper bounded by the sum of the local alternating regrets of the nodes lying in the path of the *best fixed leaf/action*.

**Definition 3.7.** For any sequence $c^1, \ldots, c^T \in [-1, 1]^n$ the *alternating local regret* of a node $s \in V$, denoted as $\mathcal{R}_{loc}^T(s)$, is defined as

$$\mathcal{R}_{loc}^T(s) := \sum_{t \in [T]} \left( q(s, \pi^t, c^t) + q(s, \pi^t, c^{t-1}) \right)^\top \pi^t(s) - \min_{\alpha \in \{\ell(s), r(s)\}} \sum_{t \in [T]} \left( Q(\alpha, \pi^t, c^t) + Q(\alpha, \pi^t, c^{t-1}) \right)$$

**Lemma 3.8.** *Let a leaf/action $i \in L$ and consider the path $p = (\text{root} = s_1, \ldots, s_H = i)$ from the root to the leaf $i \in L$. Then, $\sum_{t=1}^T (c^t + c^{t-1})^\top x^{\pi^t} - 2\sum_{t=1}^T c_i^t \leq \sum_{\ell=1}^H \mathcal{R}_{loc}(s_\ell)$.*

Up to this point, it is evident that in order to bound the overall alternating regret of Algorithm 3, we just need to bound the local alternating regret of any node $s \in V$. Using Theorem 3.4 we can bound the local regret of leaves $i \in L$ for which $q(i, \pi^t, c^{t-1}) = q(i, \pi^{t-1}, c^{t-1})$. However this approach does apply for nodes $s \in V/L$ since the local regret does not have the *alternating structure*, $q(s, \pi^t, c^{t-1}) \neq q(s, \pi^{t-1}, c^{t-1})$. To overcome the latter in Lemma 3.9 we establish that $q(s, \pi^t, c^{t-1}), q(s, \pi^{t-1}, c^{t-1})$ are in distance $\mathcal{O}(\gamma)$ which permits us to bound $\mathcal{R}_{loc}^T(s)$ for $s \in V/L$ by tweaking the proof of Theorem 3.4.

**Lemma 3.9.** *Let $\pi^1, \ldots, \pi^T$ the policies produced by Algorithm 3 then for any node $s \in V$, i) $\|\pi^t(s) - \pi^{t-1}(s)\|_1 \leq 48\gamma$ and ii) $\|q(s, \pi^t, c^{t-1}) - q(s, \pi^{t-1}, c^{t-1})\|_\infty \leq 48\gamma \log n$.*

Using Lemma 3.9 we can establish an upper bound on the local regret of any actions $s \in V$. The proof of Lemma 3.10 lies in Appendix B and follows a similar structure with the proof of Theorem 3.4.

**Lemma 3.10.** *Let $\gamma := \mathcal{O}\left( \log^{1/3} T / (T^{1/3} \log^{1/3} n) \right)$ in Algorithm 3 then $\mathcal{R}_{loc}^T(s) \leq \mathcal{O}\left( \log^{2/3} T \cdot \log^{1/3} n \cdot T^{1/3} \right)$ for all $s \in V$.*

Theorem 2.3 directly follows by combining Lemma 3.10, Lemma 3.8 and $H = \log n$.

# 4 The Ball case

In Algorithm 4 we present an online learning algorithm with $\mathcal{O}(\log T)$ for $\mathcal{D} = \mathbb{B}(0, 1)$ and $\|c^t\|_2 \leq 1$. Then through the transformation $\hat{x}_t := c + \rho x^t$ with $x^t \in \mathbb{B}(0, 1)$, Algorithm 4 can be transformed to a $\mathcal{O}(\rho \log T)$-alternating regret algorithm for $\mathcal{D} = \mathbb{B}(c, \rho)$.

Algorithm 4 may seem complicated at the first sight however it is composed by two basic algorithmic primitives. At Step 4 Algorithm 4 computes the output $w_t \in \mathcal{B}(0, 1)$ of the *Follow the Regularized Leader* (FTRL) with Euclidean regularization and adaptive step-size $r_{0:t-1}$ (Step 3 of Algorithm 4).At Step 5, it *mixes* the output $w_t \in \mathcal{B}(0, 1)$ of FTRL with the *unnormalized best-response* $-c^{t-1} \in \mathcal{B}(0, 1)$. The selection of the *mixing coefficient $p_t$* is adaptively updated at Step 7.

## 4.1 Proof of Theorem 2.4

In this section we present the main steps of the proof of Theorem 2.4. In Lemma 4.1 we provide a first upper bound on the alternating regret of Adaptive FTRL.

**Lemma 4.1.** *Let $w_1, \ldots, w_T \in \mathbb{B}(0, 1)$ the sequence produced by Adaptive FTRL (Step 4 of Algorithm 4) given as input the cost-vector sequence $c^1, \ldots, c^T \in \mathbb{B}(0, 1)$. Let $t_1$ denote the maximum time-index such that $\sum_{s=1}^t (c^s + c^{s-1})^\top w_t \geq -\sum_{s=1}^t \|c^s + c^{s-1}\|_2^2/4$. Then,*

$$\sum_{t=1}^T (c^t + c^{t-1})^\top w_t - \min_{x \in \mathbb{B}(0,1)} \sum_{t=1}^T (c^t + c^{t-1})^\top x \leq 4\sqrt{1 + \sum_{t=1}^{t_1} \|c^t + c^{t-1}\|_2^2} + \mathcal{O}(\log T)$$

Lemma 4.1 guarantees that Adaptive FTRL admits only $o(\sqrt{T})$ alternating regret in case $t_1 = o(T)$. Using Lemma 4.1, we establish Lemma 4.2 which is the cornerstone of our algorithm and guarantees that once Adaptive FTRL is *appropriately* mixed with unnormalized best-response $(-c^{t-1})$, then the resulting algorithm always admits $\mathcal{O}(\log T)$ regret.

**Lemma 4.2.** *Let $w_1, \ldots, w_T \in \mathbb{B}(0, 1)$ be produced by Adaptive FTRL given as input $c^1, \ldots, c^T \in \mathbb{B}(0, 1)$ and $t_1$ be the maximum round such that $\sum_{s=1}^t (c^s + c^{s-1})^\top w_s \geq -\sum_{s=1}^t \|c^s + c^{s-1}\|_2^2/4$.*

---

**Algorithm 4** Online Learning Algorithm for Unit Ball

---

1: $p_1 \leftarrow 0$, $D_1 \leftarrow [0,1]$ and $c^0 \leftarrow (0,\ldots,0)$.
2: **for** each round $t = 1, \cdots, T$ **do**
3:     The learner computes the coefficient $r_{0:t-1} \leftarrow \sqrt{1 + \sum_{s=1}^{t-1} \|c^s + c^{s-1}\|_2^2}$
4:     The learner computes the output of FTRL,

$$w_t \leftarrow \operatorname{argmin}_{\|x\| \leq 1} \left[ \sum_{s=1}^{t-1} (c^s + c^{s-1})^\top x + \frac{r_{0:t-1}}{2} \|x\|_2^2 \right] \quad \text{\# Adaptive FTRL}$$

5:     The learner **selects** the action $x^t \leftarrow (1 - p_t) w_t + p_t(-c^{t-1})$     # Mixing Adaptive FTRL
    with Unormalized Best-Response
6:     The adversary **selects** cost $c^t$ with $\|c^t\|_2 \leq 1$ and the learner **suffers** cost $(c^{t-1} + c^t)^\top x^t$.
7:     The learner updates the interval $D_t \subseteq [0,1]$ as follows,

$$D_t \leftarrow \left[ 0, \min\left( 1, \frac{20}{\sqrt{1 + \sum_{s=1}^{t} \|c^s + c^{s-1}\|_2^2}} \right) \right]$$

    and then updates the coefficient $p_t \in [0,1]$ as follows,

$$p_{t+1} \leftarrow \left[ p_t + \frac{20(c^t + c^{t-1})^\top \cdot (x^t + c^{t-1})}{1 + \sum_{s=1}^{t} \|c^s + c^{s-1}\|_2^2} \right]_{D_t}$$

8: **end for**

---

*Let $p := 20/\sqrt{400 + \sum_{t=1}^{t_1} \|c^t + c^{t-1}\|_2^2}$ and let $y_t := (1-p)w_t - pc^{t-1}$ for $t \leq t_1$ and $y_t := w_t$ for $t \geq t_1 + 1$. Then $\sum_{t=1}^{T} (c^t + c^{t-1})^\top y_t - \min_{x \in \mathbb{B}(0,1)} \sum_{t=1}^{T} (c^t + c^{t-1})^\top x \leq \mathcal{O}(\log T)$.*

Lemma 4.2 establishes that in case at Step 5, Algorithm 4 mixed the output $w_t$ of Adaptive FTRL with the unnormalized best-response ($-c^{t-1} \in \mathcal{B}(0,1)$) as follows,

$$y_t := (1 - q_t) \cdot w_t + q_t \cdot (-c^{t-1}) \text{ with } q_t := \frac{20 \mathrm{I}\,[t \leq t_1]}{\sqrt{400 + \sum_{t=1}^{t_1} \|c^t + c^{t-1}\|_2^2}}, \tag{3}$$

then it would admit $\mathcal{O}(\log T)$ alternating regret. Obviously, Algorithm 4 *does not know a-priori* neither $t_1$ nor $\sum_{t=1}^{t_1} \|c^t + c^{t-1}\|_2^2$. However by using the recent result of [5] for *Online Gradient Descent in Shrinking Domains*, we can establish that the *mixing coefficients* $p_t \in [0,1]$ selected by Algorithm 4 at Step 7, admit the exact same result as selecting $q_t \in [0,1]$ described in Equation 3. The latter is formalized in Lemma 4.3.

**Lemma 4.3.** *Let the sequences $w_1, \ldots, w_T \in \mathbb{B}(0,1)$ and $p_1, \ldots, p_T \in (0,1)$ produced by Algorithm 4 given as input $c^1, \ldots, c^T \in \mathcal{B}(0,1)$. Additionally let $t_1$ denote the maximum time such that $\sum_{s=1}^{t} (c^s + c^{s-1})^\top w_s \geq -\sum_{s=1}^{t} \|c^s + c^{s-1}\|_2^2/4$ and consider the sequence $q_t := \mathrm{I}\,[t \leq t_1] \cdot \left( 20/\sqrt{400 + \sum_{t=1}^{t_1} \|c^t + c^{t-1}\|_2^2} \right)$. Then,*

$$\sum_{t \in [T]} (c^{t-1} + c^t)^\top (w_t + c^{t-1}) \cdot q_t - \sum_{t \in [T]} (c^{t-1} + c^t)^\top (w_t + c^{t-1}) \cdot p_t \leq \mathcal{O}(\log T)$$

## 5 Conclusion

In this paper we introduced a variant of the Online Linear Optimization that we call Alternating Online Linear Optimization for which we developed the first online learning algoithms with $o(\sqrt{T})$ regret guarantees. Our work is motivated by the popular setting of alternating play in two-player games and raises some interesting open questions. The most natural ones is understanding whether

$\tilde{\mathcal{O}}(1)$ regret guarantees can be established the $n$-dimensional simplex as well as establishing $o(\sqrt{T})$ for general convex losses.

**Limitations:** The current work is limited to the linear losses setting. Notice that the classic reduction from convex to linear losses in Standard OLM no longer holds in Alternating OLM. Therefore the generalization to general convex losses seems to require new techniques. We defer this study for future work.

## Acknowledgements

This work was supported by the Swiss National Science Foundation (SNSF) under grant number 200021_205011, by Hasler Foundation Program: Hasler Responsible AI (project number 21043) and Innovation project supported by Innosuisse (contract agreement 100.960 IP-ICT). Luca Viano is funded through a PhD fellowship of the Swiss Data Science Center, a joint venture between EPFL and ETH Zurich. This research was also supported in part by the National Research Foundation, Singapore and DSO National Laboratories under its AI Singapore Program (AISG Award No: AISG2-RP-2020-016), grant PIESGP-AI-2020-01, AME Programmatic Fund (Grant No.A20H6b0151) from A*STAR and Provost's Chair Professorship grant RGEPPV2101.

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

# A Omitted Proofs of Section 3

## A.1 Auxilliary Lemmas

**Lemma A.1.** *The log-barrier function $R(x) = -\log x - \log(1-x)$ is 1-strongly convex in $[0,1]$. More precisely, for all $x, y \in [0,1]$*

$$R(y) \geq R(x) + R'(x)^\top (y-x) + \frac{1}{2}|x-y|^2$$

*Proof.* Let $f(x) := -\log x$ then $f'(x) = -\frac{1}{x}$ and $f''(x) = \frac{1}{x^2}$. Since $x \leq 1$ we get that $f''(x) \geq 1$ and thus

$$f(y) \geq f(x) + f'(x)(y-x) + \frac{1}{2}(x-y)^2$$

At the same time the function $f(x) = -\log(1-x)$ is convex in $[0,1]$. This concludes the proof. $\square$

**Lemma A.2.** *Let $x := \operatorname{argmin}_{z \in [0,1]} [\gamma c \cdot z + R(z)]$ and $y := \operatorname{argmin}_{z \in [0,1]} [\gamma \hat{c} \cdot z + R(z)]$ where $R(\cdot)$ is an 1-strongly convex function in $\mathbb{R}$. Then,*

$$|x-y| \leq 2\gamma |c - \hat{c}|$$

*Proof.* By the strong convexity of the function $\gamma c^T z + R(z)$ and first order optimality conditions for $x$, we get that

$$\gamma c^\top y + R(y) \quad \geq \quad \gamma c^\top x + R(x) + \frac{1}{2}|x-y|^2$$

As a result, we get that

$$
\begin{aligned}
\frac{1}{2}|x-y|^2 \quad &\leq \quad \gamma c \cdot (y-x) + R(y) - R(x) \\
&= \quad \gamma \hat{c} \cdot (y-x) + \gamma(c - \hat{c}) \cdot (y-x) + R(y) - R(x) \\
&\leq \quad \gamma(c - \hat{c}) \cdot (y-x)
\end{aligned}
$$

which implies that $|x-y| \leq 2\gamma |c - \hat{c}|$. $\square$

## A.2 Proof of Lemma 3.2

**Lemma 3.2.** *Let $y^1, \ldots, y^T \in \Delta_2$ where $y^t := \min_{x \in \Delta_2} \left[ (c^t + c^{t-1})^\top x + \sum_{s=1}^{t-1} (c^s + c^{s-1})^\top x + R(x)/\gamma \right]$. Then, $\sum_{t=1}^T (c^t + c^{t-1})^\top x^t - \min_{i \in [2]} \sum_{t=1}^T (c_i^t + c_i^{t-1}) \leq 2\log T/\gamma + 2$.*

*Proof.* We start by rewrite the regret minimization problem over $\Delta_2$ as an equivalent one over $[0,1]$, that is

$$\sum_{t=1}^T (c^t + c^{t-1})^\top (x^t - x^\star) = \sum_{t=1}^T (\hat{c}^t + \hat{c}^{t-1})^\top (x_1^t - x_1^\star)$$

where $\hat{c}^t = c_1^t - c_2^t$. Moreover notice that

$$y_1^t := \operatorname*{arg\,min}_{p \in [0,1]} \left[ \sum_{\tau=1}^t (\hat{c}^\tau + \hat{c}^{\tau-1})p - \frac{\log p + \log(1-p)}{\gamma} \right] \tag{4}$$

By the "Follow the Leader/Be the Leader" Lemma [7, Lemma 3.1] , we have that

$$\left[ \sum_{t=1}^T (\hat{c}^t + \hat{c}^{t-1})y_1^t - \frac{\log y_1^t + \log(1-y_1^t)}{\gamma} \right] \leq \min_{p \in [0,1]} \left[ \sum_{t=1}^T (\hat{c}^t + \hat{c}^{t-1})p - \frac{\log p + \log(1-p)}{\gamma} \right].$$

That implies

$$\left[ \sum_{t=1}^T (\hat{c}^t + \hat{c}^{t-1})y_1^t \right] \leq \min_{p \in [0,1]} \left[ \sum_{t=1}^T (\hat{c}^t + \hat{c}^{t-1})p - \frac{\log p + \log(1-p)}{\gamma} \right]$$

Now let $x^\star = \arg\min_{p \in [0,1]} \sum_{t=1}^{T}(\hat{c}^t + \hat{c}^{t-1})p$ and let us subtract $\sum_{t=1}^{T}(\hat{c}^t + \hat{c}^{t-1})x^\star$ from both sides

$$\left[\sum_{t=1}^{T}(\hat{c}^t + \hat{c}^{t-1})(y_1^t - x^\star)\right] \leq \min_{p \in [0,1]}\left[\sum_{t=1}^{T}(\hat{c}^t + \hat{c}^{t-1})p - \frac{\log p + \log(1-p)}{\gamma}\right] - \sum_{t=1}^{T}(\hat{c}^t + \hat{c}^{t-1})x^\star$$

In case $x^\star = 0$, we upper bound the minimum on the right hand side with the same expression evaluated at $p := 1/T$. As a result,

$$\left[\sum_{t=1}^{T}(\hat{c}^t + \hat{c}^{t-1})(y_1^t - 0)\right] \leq \sum_{t=1}^{T}(\hat{c}^t + \hat{c}^{t-1})\frac{1}{T} - \frac{\log(\frac{1}{T}) + \log(1 - \frac{1}{T})}{\gamma}$$

$$\leq 2 + \frac{\log(T) + \log(\frac{T}{T-1})}{\gamma} \leq 2 + \frac{2\log T}{\gamma} \tag{5}$$

In case $x^\star = 1$, we upper bound the minimum on the right hand side by the expression evaluated at $p := 1 - 1/T$. As a result,

$$\left[\sum_{t=1}^{T}(\hat{c}^t + \hat{c}^{t-1})(y_1^t - 1)\right] \leq \sum_{t=1}^{T}(\hat{c}^t + \hat{c}^{t-1})\left(1 - \frac{1}{T}\right) - \frac{\log(\frac{1}{T}) + \log(1 - \frac{1}{T})}{\gamma} - \sum_{t=1}^{T}(\hat{c}^t + \hat{c}^{t-1})$$

$$\leq 2 + \frac{\log(T) + \log(\frac{T}{T-1})}{\gamma} \leq 2 + \frac{2\log T}{\gamma} \tag{6}$$

Therefore putting together Equation (5) and Equation (6), we can conclude that $\sum_{t=1}^{T}(c^t + c^{t-1})^\top x^t - \min_{i \in [2]} \sum_{t=1}^{T}(c_i^t + c_i^{t-1}) \leq 2\log T/\gamma + 2$.

$\square$

### A.3  Proof of Lemma 3.3

Before presenting the formal proof of Lemma 3.3 we present Lemma A.3 and Lemma A.4 that are necessary for its proof.

**Lemma A.3.** *Let $x^t$ as in Algorithm 2 and $y_1^t$ be the BTRL update as in Lemma 3.2 with $R(x) = -\log x - \log(1-x)$ and $x_1^t$ be the update as in Algorithm 3. Then the following hold.*

- $|x_1^t - y_1^t| \leq 8\gamma$

- $|x_1^t - x_1^{t+1}| \leq 16\gamma$

- $\left|y_1^t - y_1^{t+1}\right| \leq 8\gamma$

*Proof.* Notice that for any $x, y \in \Delta_2$ and cost vector $c = (c_1, c_2) \in \mathbb{R}^2$, we have that

$$c^\top(x - y) = c_1(x_1 - y_1) + c_2(x_2 - y_2) = c_1(x_1 - y_1) + c_2(-x_1 + y_1) = (x_1 - y_1)(c_1 - c_2).$$

This means that we can reduce the bidimensional update in Algorithm 2 as

$$x_1^t = \arg\min_{p \in [0,1]}\left[2(c_1^{t-1} - c_2^{t-1})p + \sum_{s=1}^{t-1}(c_1^s - c_2^s + c_1^{s-1} - c_2^{s-1})p - \frac{\log p + \log(1-p)}{\gamma}\right] \tag{7}$$

At this point, using strong convexity of the log barrier function (Lemma A.1), the form of the updates in Equation (4) and Equation (7), we can invoke Lemma A.2 using $x = x_1^t$ and $y = y_1^t$, this gives

$$\left|x_1^t - y_1^t\right| \leq 2\gamma\left|2c_1^{t-1} - 2c_2^{t-1} - c_1^t - c_1^{t-1} + c_2^t + c_2^{t-1}\right| \leq 8\gamma$$

where we used that the cost sequence is in $[-1, 1]$. For the second fact, we invoke again Lemma A.2 but with $x = x_1^t$ and $y = x_1^{t+1}$ and we obtain

$$\left|x_1^t - x_1^{t+1}\right| \leq 2\gamma\left|2c_1^{t-1} - 2c_2^{t-1} - c_1^t - c_1^{t-1} + c_2^t + c_2^{t-1} - 2c_1^t + 2c_2^t\right| \leq 16\gamma$$

For the third fact, we use Lemma A.2 but with $x = y_1^t$ and $y = y_1^{t+1}$ and we obtain

$$\left|y_1^t - y_1^{t+1}\right| \leq 2\gamma\left|c_1^{t+1} - c_2^{t+1} - c_1^t + c_2^t\right| \leq 8\gamma$$

$\square$

**Lemma A.4.** *Let $(x, y) \in [0, 1]^2$ and $(x', y') \in [0, 1]^2$ such that $|x - y| \le B$ , $|x - y'| \le B$, $|x' - y| \le B$ and $|x' - y'| \le B$ with $B \le \frac{1}{8}$ then*

$$|A^{-1}(x, y) - A^{-1}(x', y')| \le 192|x - x'| + 192|y - y'|$$

*where $A(x, y) = (xy)^{-1} - (1 - x)^{-1}(1 - y)^{-1}$.*

*Proof.* To simplify notation we denote $x_t := tx + (1 - t)x'$ and $y_t := ty + (1 - t)y'$. Then

$$A^{-1}(x, y) - A^{-1}(x', y') = \int_0^1 \langle \nabla A^{-1}(x_t, y_t), (x, y) - (x', y') \rangle \, \partial t$$

$$\le \max_{t \in [0, 1]} \|\nabla A^{-1}(x_t, y_t)\|_\infty \cdot \|(x, y) - (x', y')\|_1 \tag{8}$$

Let us focus on bounding $\|\nabla A^{-1}(x_t, y_t)\|_\infty$. Notice that

$$\left| \frac{\partial A^{-1}(x_t, y_t)}{\partial x} \right| \le \frac{3}{((1 - x_t)(1 - y_t) + x_t y_t)^2}. \tag{9}$$

Now, notice that $|x_t - y_t| \le t|x - y| + (1 - t)|x' - y'| \le B$. Using the latter we can lower bound the denominator of Equation 9. More precisely,

$$(1 - x_t)(1 - y_t) + x_t y_t = x_t^2 + (1 - x_t)^2 + (2x_t - 1)(y_t - x_t)$$

$$\ge \frac{1}{4} - |2x_t - 1| \, |y_t - x_t|$$

$$\ge \frac{1}{4} - B$$

So for $B \le \frac{1}{8}$ we obtain

$$\left| \frac{\partial A^{-1}(x_t, y_t)}{\partial x} \right| \le 3 \cdot 8^2 = 192.$$

By symmetricity, we can bound with analogous steps the partial derivative wrt to $y$ and hence we get

$$\|\nabla A^{-1}(x_t, y_t)\|_\infty \le 192.$$

Plugging this bound back in Equation (8) concludes the proof. $\qquad \square$

**Lemma 3.3.** Let $x^t = (x_1^t, x_2^t) \in \Delta_2$ as in Algorithm 2 and $y^t = (y_1^t, y_2^t) \in \Delta_2$ as in Lemma 3.2. Then,

$$x_1^t - y_1^t = \gamma A^{-1}(x_1^t, y_1^t) \cdot \left( (c_1^t - c_2^t) - (c_1^{t-1} - c_2^{t-1}) \right)$$

with $A(x_1, y_1) := (x_1 y_1)^{-1} + (1 - x_1)^{-1}(1 - y_1)^{-1}$ and $|A^{-1}(x_1^t, y_1^t) - A^{-1}(x_1^{t+1}, y_1^{t+1})| \le \mathcal{O}(\gamma)$.

*Proof.* In order to prove this Lemma 3.3, we use the equivalent one-dimensional description provided in Equation 10.

$$x_1^t = \arg\min_{p \in [0, 1]} \left[ 2(c_1^{t-1} - c_2^{t-1})p + \sum_{s=1}^{t-1}(c_1^s - c_2^s + c_1^{s-1} - c_2^{s-1})p - \frac{\log p + \log(1 - p)}{\gamma} \right]. \tag{10}$$

Similarly the update of BTRL in Lemma 3.2 can be equivalently descirbed as,

$$y_1^t = \arg\min_{p \in [0, 1]} \left[ \sum_{s=1}^{t}(c_1^s - c_2^s + c_1^{s-1} - c_2^{s-1})p - \frac{\log p + \log(1 - p)}{\gamma} \right]. \tag{11}$$

Since $\lim_{p \to \partial[0,1]} R(p) = \infty$ both $x_1^t, y_1^t \in [0, 1] \setminus \partial[0, 1]$. Therefore, the first order optimality for Equation (7) requires that

$$2\gamma(c_1^{t-1} - c_2^{t-1}) + \gamma \sum_{s=1}^{t-1} c_1^s + c_1^{s-1} - (c_2^s + c_2^{s-1}) - \frac{1}{x_1^t} + \frac{1}{1 - x_1^t} = 0 \tag{12}$$

Using the same reasoning for the BTRL updates in Equation (4)

$$\gamma(c_1^t - c_2^t) + \gamma(c_1^{t-1} - c_2^{t-1}) + \gamma\sum_{s=1}^{t-1}(c_1^s + c_1^{s-1}) - (c_2^s + c_2^{s-1}) - \frac{1}{y_1^t} + \frac{1}{1-y_1^t} = 0. \qquad (13)$$

Now, subtracting Equation (12) to Equation (13), we obtain

$$\gamma(c_1^t - c_2^t - c_1^{t-1} + c_2^{t-1}) - \frac{1}{y_1^t} + \frac{1}{x_1^t} + \frac{1}{1-y_1^t} - \frac{1}{1-x_1^t} = 0$$

that implies

$$\gamma(c_1^t - c_2^t) - \gamma(c_1^{t-1} - c_2^{t-1}) = (x_1^t - y_1^t)\underbrace{\left(\frac{1}{x_1^t y_1^t} + \frac{1}{(1-y_1^t)(1-x_1^t)}\right)}_{A(x_1^t, y_1^t)}.$$

Therefore, we can express the difference between the updates as a function of the costs according to the following formula

$$x_1^t - y_1^t = \gamma A^{-1}(x_1^t, y_1^t)\left((c_1^t - c_2^t) - (c_1^{t-1} - c_2^{t-1})\right). \qquad (14)$$

We conclude the proof by establishing that

$$\left|A^{-1}(x_1^t, y_1^t) - A^{-1}(x_1^{t+1}, y_1^{t+1})\right| \le \mathcal{O}(\gamma).$$

By Lemma A.3 we are ensured that

- $|x_1^t - y_1^t| \le 8\gamma$
- $|x_1^t - x_1^{t+1}| \le 16\gamma$
- $\left|y_1^t - y_1^{t+1}\right| \le 8\gamma$

In case $\gamma \le 1/(16 \cdot 8)$ we are ensured that the conditions of Lemma A.4 are satisfied ($B \le 1/8$) and thus

$$\left|A^{-1}(x_1^t, y_1^t) - A^{-1}(x_1^{t+1}, y_1^{t+1})\right| \le 192(\left|x_1^t - x_1^{t+1}\right| + \left|y_1^t - y_1^{t+1}\right|)$$

Combining the latter with the guarantees of Lemma A.4 we get that

$$\left|A^{-1}(x_1^t, y_1^t) - A^{-1}(x_1^{t+1}, y_1^{t+1})\right| \le 192 \cdot 24\gamma$$

$\square$

# B Omitted proofs for the $n$ dimensional case.

## B.1 Auxiliary Lemmas

**Corollary B.1.** $i)$ $Q(s, \pi, c) = q(s, \pi, c)^\top \cdot \pi(s)$ $ii)$ $c^\top x^\pi = Q(\text{root}, \pi, c)$.

*Proof.* For fact i) for any $s \in \text{Level}(h)$, we have that

$$
\begin{aligned}
Q(s, \pi, c) &= \sum_{i \in L} \Pr(s, i, \pi) c_i \\
&= \sum_{i \in L} \pi(\ell(s)|s) \Pr(\ell(s), i, \pi) c_i + \sum_{i \in L} \pi(r(s)|s) \Pr(r(s), i, \pi) c_i \\
&= \pi(\ell(s)|s) \sum_{h \in L} \Pr(\ell(s), i, \pi) c_i + \pi(r(s)|s) \sum_{i \in L} \Pr(r(s), i, \pi) c_i \\
&= \pi(\ell(s)|s) Q(\ell(s), \pi, c) + \pi(r(s)|s) Q(r(s), \pi, c) \\
&= q(s, \pi, c)^\top \cdot \pi(s)
\end{aligned}
$$

where the second last equality uses the fact that $s \in \text{Level}(h) \implies \ell(s), r(s) \in \text{Level}(h+1)$.

Finally, fact ii) follows trivially from the definition of $x^\pi$. Indeed, we have that

$$
c^\top \cdot x^\pi = \sum_{i \in L} x^\pi(i) c_i = \sum_{i \in L} \Pr(\text{root}, i, \pi) c_i = Q(\text{root}, \pi, c)
$$

$\square$

## B.2 Proof of Lemma 3.8

**Lemma 3.8.** Let a leaf node $i \in L$ and let $p = (\text{root} = s_1, \ldots, s_H = i)$ denotes the path from the root to $i$. Then the following holds,

$$
\sum_{t=1}^T (c^t + c^{t-1})^\top \cdot x^{\pi^t} - 2 \sum_{t=1}^T c_i^t \leq \sum_{s_\ell \in p} \mathcal{R}_{\text{loc}}(s_\ell)
$$

*Proof.* By Item 2 of Corollary B.1 and the fact that $c^0 = 0$, we get

$$
\begin{aligned}
\sum_{t=1}^T (c^t + c^{t-1})^\top \cdot x^{\pi^t} - 2 \sum_{t=1}^T c_i^t &= \sum_{t=1}^T \left( Q(\text{root}, \pi^t, c^t) + Q(\text{root}, \pi^t, c^{t-1}) - Q(i, \pi^t, c^t) - Q(i, \pi^t, c^{t-1}) \right) \\
&= \sum_{t=1}^T \sum_{\ell=1}^{H-1} \left( Q(s_\ell, \pi^t, c^t) + Q(s_\ell, \pi^t, c^{t-1}) - Q(s_{\ell+1}, \pi^t, c^t) - Q(s_{\ell+1}, \pi^t, c^{t-1}) \right) \\
&= \sum_{t=1}^T \sum_{\ell=1}^{H-1} \left( Q(s_\ell, \pi^t, c^t) + Q(s_\ell, \pi^t, c^{t-1}) \right) \\
&\quad - \min_{\alpha \in \{\ell(s_\ell), r(s_\ell)\}} \sum_{t=1}^T \left( Q(\alpha, \pi^t, c^t) + Q(\alpha, \pi^t, c^{t-1}) \right) \\
&= \sum_{t=1}^T \sum_{\ell=1}^{H-1} \left( q(s_\ell, \pi^t, c^t) + q(s_\ell, \pi^t, c^{t-1}) \right)^\top \cdot \pi^t(s_\ell) \\
&\quad - \min_{\alpha \in \{\ell(s_\ell), r(s_\ell)\}} \sum_{t=1}^T \left( Q(\alpha, \pi^t, c^t) + Q(\alpha, \pi^t, c^{t-1}) \right) \quad \text{Corollary B.1} \\
&= \sum_{s_\ell \in p} \mathcal{R}_{\text{loc}}(s_\ell)
\end{aligned}
$$

$\square$

## B.3 Proof of Lemma 3.9

**Lemma 3.9.** Let $\pi^1, \ldots, \pi^T$ the policies produced by Algorithm 3 then for any state $s \in V$, $i)$ $\|\pi^t(s) - \pi^{t-1}(s)\|_1 \le 48\gamma$ and $ii)$ $\|q(s, \pi^t, c^{t-1}) - q(s, \pi^{t-1}, c^{t-1})\|_\infty \le 48\gamma \log n$.

*Proof.* We first establish that $\|\pi^t(s) - \pi^{t-1}(s)\|_1 \le 48\gamma$.

Let $\bar{Q}(s, \pi, c) := Q(\ell(s), \pi, c) - Q(r(s), \pi, c)$ then policy update in Step 6 of Algorithm 3 admits the following one dimensional form,

$$\pi^t(\ell(s)|s) = \arg\min_{x \in [0,1]} \left[ 2\gamma(\bar{Q}(s, \pi^t, c^t) + \bar{Q}(s, \pi^t, c^{t-1})) + \gamma \sum_{\tau=1}^{t-1}(\bar{Q}(s, \pi^\tau, c^\tau) + \bar{Q}(s, \pi^\tau, c^{\tau-1})) + R(x) \right].$$

Similarly for the policy $\pi^{t-1}$,

$$\pi^{t-1}(\ell(s)|s) = \arg\min_{x \in [0,1]} \left[ 2\gamma(\bar{Q}(s, \pi^{t-1}, c^{t-1}) + \bar{Q}(s, \pi^{t-1}, c^{t-2})) + \gamma \sum_{\tau=1}^{t-2}(\bar{Q}(s, \pi^\tau, c^\tau) + \bar{Q}(s, \pi^\tau, c^{\tau-1})) + R(x) \right].$$

Using Lemma A.2 we get that,

$$\left| \pi^t(\ell(s)|s) - \pi^{t-1}(\ell(s)|s) \right| \le 2\gamma \Big| 2\bar{Q}(s, \pi^t, c^t) + 2\bar{Q}(s, \pi^t, c^{t-1}) + \bar{Q}(s, \pi^{t-1}, c^{t-1}) + \bar{Q}(s, \pi^{t-1}, c^{t-2})$$

$$- 2\bar{Q}(s, \pi^{t-1}, c^{t-1}) - 2\bar{Q}(s, \pi^{t-1}, c^{t-2}) \Big| \le 24\gamma$$

where the last inequality comes from the fact that $-1 \le Q(s, \pi, c) \le 1$ and thus $\left| \bar{Q}(s, \pi, c) \right| \le 2$.

Up next we establish that

$$\|q(s, \pi^t, c^{t-1}) - q(s, \pi^{t-1}, c^{t-1})\|_\infty \le 48\gamma \log n.$$

To simplify notation we let $h_0$ denote the depth of state $s_0 \in V$. Up next we prove that

$$\|q(s_0, \pi^t, c^{t-1}) - q(s_0, \pi^{t-1}, c^{t-1})\|_\infty \le 48\gamma \log n.$$

In order to prove the latter we deploy a *coupling argument* by considering two correlated random walks $(s_0, s_0), (s_1, s'_1), \ldots, (s_H, s'_H)$ where both walks are initialized at $(s_0, s_0)$ while at each level $h \in \{h_0, \ldots, H-1\}$, the first walk marginally follows policy $\pi \in \Delta_2$ while the second walk marginally follows $\pi' \in \Delta_2$.

More precisely, let $(s_h, s'_h)$ the pair of nodes visited respectively by the first and the second walk at level $h \in \{h_0, \ldots, H-1\}$. Then the next pair of nodes $(s_h, s'_h)$ follow the following joint probability distribution.

- **In case $s'_h \ne s_h$:** The next pair of nodes $(s_{h+1}, s'_{h+1})$ are independent random variables respectively following $\pi(s_h) \in \Delta_2$ and $\pi'(s'_h) \in \Delta_2$. More precisely,

$$s_{h+1} = \begin{cases} \ell(s_h) & \text{w.p.} \quad \pi(\ell(s_h)|s_h) \\ r(s_h) & \text{w.p.} \quad 1 - \pi(\ell(s_h)|s_h) \end{cases} \quad \text{and} \quad s'_{h+1} = \begin{cases} \ell(s'_h) & \text{w.p.} \quad \pi'(\ell(s'_h)|s'_h) \\ r(s'_h) & \text{w.p.} \quad 1 - \pi'(\ell(s'_h)|s'_h) \end{cases}$$

- **In case $s_h = s'_h = s$ and $\pi(\ell(s)|s) \le \pi'(\ell(s)|s)$:** Then the next pair of nodes $(s_j, s'_h)$ follows the joint probability distribution,

$$(s_{h+1}, s'_{h+1}) = \begin{cases} (\ell(s), \ell(s)) & \text{w.p.} \quad \pi(\ell(s)|s) \\ (r(s), \ell(s)) & \text{w.p.} \quad \pi'(\ell(s)|s) - \pi(\ell(s)|s) \\ (r(s), r(s)) & \text{w.p.} \quad 1 - \pi'(\ell(s)|s) \end{cases}$$

- **In case $s_h = s'_h = s$ and $\pi(\ell(s)|s) \ge \pi'(\ell(s)|s)$:** Then the next pair of nodes $(s_j, s'_h)$ follows the joint probability distribution,

$$(s_{h+1}, s'_{h+1}) = \begin{cases} (\ell(s), \ell(s)) & \text{w.p.} \quad \pi'(\ell(s)|s) \\ (\ell(s), r(s)) & \text{w.p.} \quad \pi(\ell(s)|s) - \pi'(\ell(s)|s) \\ (r(s), r(s)) & \text{w.p.} \quad 1 - \pi(\ell(s)|s) \end{cases}$$

The above joint random walk, guarantees that the first random walk (resp. the second) follows policy $\pi$ (resp. $\pi'$ for the second coordinate). More precisely,

$$\Pr\left[s_{h+1} = \ell(s) \mid s_h = s\right] = \pi(\ell(s)|s) \text{ and } \Pr\left[s'_{h+1} = \ell(s) \mid s'_h = s\right] = \pi'(\ell(s)|s)$$

As a result,

$$\mathrm{E}\left[c_i - c_{i'}\right] = Q(s_0, \pi, c) - Q(s_0, \pi', c)$$

where $(i, i') \in L \times L$ denotes the pair of leaves reached by the joint random walk initialized at $s_0 \in V/L$.

$$|Q(s_0, \pi, c) - Q(s_0, \pi', c)| = |\mathrm{E}\left[c_i - c_{i'}\right]| \le \mathrm{E}\left[|c_i - c_{i'}|\right]$$

$$= \sum_{h=h_0}^{H-1} \sum_{s \in \mathrm{Level}(h)} \mathrm{E}\left[|c_i - c_{i'}| \, | \, s'_{h+1} \neq s_{h+1}, s'_h = s_h = s\right] \mathrm{P}\left[s'_{h+1} \neq s_{h+1}, s'_h = s_h = s\right]$$

$$\le 2 \sum_{h=h_0}^{H-1} \sum_{s \in \mathrm{Level}(h)} \mathrm{P}\left[s'_{h+1} \neq s_{h+1}, s'_h = s_h = s\right]$$

$$= 2 \sum_{h=h_0}^{H-1} \sum_{s \in \mathrm{Level}(h)} \mathrm{P}\left[s'_{h+1} \neq s_{h+1}|s'_h = s_h = s\right] \mathrm{P}\left[s'_h = s_h = s\right]$$

$$\le 2 \sum_{h=h_0}^{H-1} \sum_{s \in \mathrm{Level}(h)} |\pi(\ell(s)|s) - \pi'(\ell(s)|s)| \, \mathrm{P}\left[s'_h = s_h = s\right]$$

where in the second equality we used the fact that $\left\{s'_{h+1} \neq s_{h+1}, s'_h = s_h = s\right\}_{s \in V, h \in [H]}$ are disjoint events.

By setting $\pi' = \pi^t$ and $\pi = \pi^{t-1}$ we get that

$$\left|Q(s_0, \pi^t, c) - Q(s_0, \pi^{t-1}, c)\right| \le 2 \sum_{h=h_0}^{H-1} \sum_{s \in \mathrm{Level}(h)} \left|\pi^t(\ell(s)|s) - \pi^{t-1}(\ell(s)|s)\right| \mathrm{P}\left[s'_h = s_h = s\right]$$

$$\le 48\gamma \sum_{h=h_0}^{H-1} \sum_{s \in \mathrm{Level}(h)} \mathrm{P}\left[s'_h = s_h = s\right]$$

$$= 48\gamma \sum_{h=h_0}^{H-1} 1 = 48\gamma \log n$$

Finally,

$$\|q(s_0, \pi^t, c^{t-1}) - q(s_0, \pi^{t-1}, c^{t-1})\|_\infty = \max_{\alpha \in \{\ell(s_0), r(s_0)\}} \left|Q(\alpha, \pi^t, c^{t-1}) - Q(\alpha, \pi^{t-1}, c^{t-1})\right| \le 48\gamma \log n.$$

$\square$

### B.4   Proof of Lemma 3.10

**Lemma 3.10.** Let $\gamma := \mathcal{O}\left(\log^{1/3} T / (T^{1/3} \log^{1/3} n)\right)$ in Algorithm 3 then $\mathcal{R}_{loc}^T(s) \le \mathcal{O}\left(\log^{2/3} T \cdot \log^{1/3} n \cdot T^{1/3}\right)$ for all $s \in V$.

*Proof.* Let the step-size $\gamma > 0$ of Algorithm 3 defined as $\gamma := \frac{1}{32 \cdot 8} \left(\frac{\log(T)}{T \log n}\right)^{\frac{1}{3}}$. Let us also introduce the BTRL update for state $s$ that is

$$\tilde{\pi}^t(s) := \arg\min_{x \in \Delta_2} \left[\sum_{\tau=1}^{t} \left(q(s, \pi^\tau, c^{\tau-1}) + q(s, \pi^\tau, c^\tau)\right)^\top x + R(x)/\gamma\right] \tag{15}$$

We can bound two separate sources of regret, according to the decomposition

$$\mathcal{R}_{loc}^T(s) = \underbrace{\sum_{t=1}^{T} \left(q(s, \pi^t, c^t) + q(s, \pi^{t-1}, c^t)\right)^{\top} \cdot \tilde{\pi}^t(s) - \min_{\alpha \in \{\ell(s), r(s)\}} \sum_{t=1}^{T} \left(Q(\alpha, \pi^t, c^t) + Q(\alpha, \pi^{t-1}, c^t)\right)}_{\text{Term I}}$$

$$+ \underbrace{\sum_{t=1}^{T} \left(q(s, \pi^t, c^t) + q(s, \pi^{t-1}, c^t)\right)^{\top} \cdot \left(\pi^t(s) - \tilde{\pi}^t(s)\right)}_{\text{Term II}} \tag{16}$$

First, we recognize that Term I is the BTRL local regret, therefore applying Lemma 3.2, we have

$$\text{Term I} \leq \mathcal{O}\left(\frac{\log T}{\gamma}\right)$$

Then, it remains to bound the term that quantifies the closeness between $\pi^t$ and $\tilde{\pi}^t$, that is

$$\sum_{t=1}^{T} \left(q(s, \pi^t, c^t) + q(s, \pi^{t-1}, c^t)\right)^{\top} \cdot \left(\pi^t(s) - \tilde{\pi}^t(s)\right)$$

Let $\bar{Q}(s, \pi, c) := Q(\ell(s), \pi, c) - Q(r(s), \pi, c)$ then by using Corollary B.1 we get that

$$\sum_{t=1}^{T} \left(q(s, \pi^t, c^t) + q(s, \pi^{t-1}, c^t)\right)^{\top} \cdot \left(\pi^t(s) - \tilde{\pi}^t(s)\right) = \sum_{t=1}^{T} \left[\bar{Q}(s, \pi^t, c^t) + \bar{Q}(s, \pi^t, c^{t-1})\right] \cdot \left[\pi^t(\ell(s)|s) - \tilde{\pi}^t(\ell(s)|s)\right] \tag{17}$$

At the same time by Lemma 3.3 we get that

$$\pi^t(\ell(s)|s) - \tilde{\pi}^t(\ell(s)|s) = \gamma \frac{\bar{Q}(s, \pi^t, c^t) - \bar{Q}(s, \pi^t, c^{t-1})}{A(\pi^t(\ell(s)|s), \tilde{\pi}^t(\ell(s)|s))} \tag{18}$$

Hence combining Equation 17 with Equation 18 we obtain

$$\sum_{t=1}^{T} \left(q(s, \pi^t, c^t) + q(s, \pi^{t-1}, c^t)\right)^{\top} \cdot \left(\pi^t(s) - \tilde{\pi}^t(s)\right) = \gamma \sum_{t=1}^{T} \frac{\bar{Q}^2(s, \pi^t, c^t) - \bar{Q}^2(s, \pi^t, c^{t-1})}{A(\pi^t(\ell(s)|s), \tilde{\pi}^t(\ell(s)|s))}$$

At this point, we notice that unfortunately we can not rearrange the sum easily because of the term $\bar{Q}^2(s, \pi^t, c^{t-1})$ that depends on both indices $t$ and $t-1$. To go around this issue, we add and subtract the term $\frac{\bar{Q}^2(s, \pi^{t-1}, c^{t-1})}{A(\pi^t(\ell(s)|s), \tilde{\pi}^t(\ell(s)|s))}$,

$$\sum_{t=1}^{T} \left(q(s, \pi^t, c^t) + q(s, \pi^{t-1}, c^t)\right)^{\top} \cdot \left(\pi^t(s) - \tilde{\pi}^t(s)\right) = \gamma \sum_{t=1}^{T} \frac{\bar{Q}^2(s, \pi^t, c^t) - \bar{Q}^2(s, \pi^{t-1}, c^{t-1})}{A(\pi^t(\ell(s)|s), \tilde{\pi}^t(\ell(s)|s))}$$

$$+ \gamma \sum_{t=1}^{T} \frac{\bar{Q}^2(s, \pi^{t-1}, c^{t-1}) - \bar{Q}^2(s, \pi^t, c^{t-1})}{A(\pi^t(\ell(s)|s), \tilde{\pi}^t(\ell(s)|s))}. \tag{19}$$

Now we bound the first term. Notice that the assumption of Lemma A.4 are satisfied with $B = 32\gamma$ and that $\gamma \leq (8 \cdot 32)^{-1}$ ensures $B \leq \frac{1}{8}$. , Therefore, rearranging the sum and invoking Lemma A.4

for $x = \pi^t$, $y = \tilde{\pi}^t$, $x' = \pi^{t+1}$, $y' = \tilde{\pi}^{t+1}$, we get

$$\gamma \sum_{t=1}^{T} \frac{\bar{Q}^2(s, \pi^t, c^t) - \bar{Q}^2(s, \pi^{t-1}, c^{t-1})}{A(\pi^t(\ell(s)|s), \tilde{\pi}^t(\ell(s)|s))} = \gamma \sum_{t=1}^{T-1} \left( \frac{\bar{Q}^2(s, \pi^t, c^t)}{A(\pi^t(\ell(s)|s), \tilde{\pi}^t(\ell(s)|s))} - \frac{\bar{Q}^2(s, \pi^t, c^t)}{A(\pi^{t+1}(\ell(s)|s), \tilde{\pi}^{t+1}(\ell(s)|s))} \right)$$

$$+ \gamma \frac{\bar{Q}^2(s, \pi^T, c^T)}{A(\pi^T(\ell(s)|s), \tilde{\pi}^T(\ell(s)|s))}$$

$$= \gamma \sum_{t=1}^{T-1} \left( \frac{1}{A(\pi^t(\ell(s)|s), \tilde{\pi}^t(\ell(s)|s))} - \frac{1}{A(\pi^{t+1}(\ell(s)|s), \tilde{\pi}^{t+1}(\ell(s)|s))} \right) \bar{Q}^2(s, \pi^t, c^t)$$

$$+ \gamma \frac{\bar{Q}^2(s, \pi^T, c^T)}{A(\pi^T(\ell(s)|s), \tilde{\pi}^T(\ell(s)|s))}$$

$$\overset{\text{Lemma A.4}}{\leq} 192\gamma \sum_{t=1}^{T-1} \bar{Q}^2(s, \pi^t, c^t) \left( \left| \pi^t(\ell(s)|s) - \pi^{t+1}(\ell(s)|s) \right| + \left| \tilde{\pi}^t(\ell(s)|s) - \tilde{\pi}^{t+1}(\ell(s)|s) \right| \right)$$

$$+ \gamma \bar{Q}^2(s, \pi^T, c^T) \left| A^{-1}(\pi^T(\ell(s)|s), \tilde{\pi}^T(\ell(s)|s)) \right|$$

$$\overset{\text{Lemma A.3}}{\leq} 192\gamma \sum_{t=1}^{T-1} \bar{Q}^2(s, \pi^t, c^t) (24\gamma + 4\gamma) + \gamma \bar{Q}^2(s, \pi^T, c^T) \left| A^{-1}(\pi^T(\ell(s)|s), \tilde{\pi}^T(\ell(s)|s)) \right|$$

$$\leq 4 \cdot 192 \cdot 28\gamma^2 T + 32\gamma$$

where in the last inequality we used $\bar{Q}^2(s, \pi^t, c^t) \leq 4 \quad \forall t$ and $A(\pi^T(\ell(s)|s), \tilde{\pi}^T(\ell(s)|s)) \geq \frac{1}{8}$. Then, for the second term in Equation (19), we use the second fact of Lemma 3.9. In more details, we have that

$$\gamma \sum_{t=1}^{T} \frac{\bar{Q}^2(s, \pi^{t-1}, c^{t-1}) - \bar{Q}^2(s, \pi^t, c^{t-1})}{A(\pi^t(\ell(s)|s), \tilde{\pi}^t(\ell(s)|s))}$$

$$= \gamma \sum_{t=1}^{T} \frac{\left( \bar{Q}(s, \pi^{t-1}, c^{t-1}) + \bar{Q}(s, \pi^t, c^{t-1}) \right) \cdot \left( \bar{Q}(s, \pi^{t-1}, c^{t-1}) - \bar{Q}(s, \pi^t, c^{t-1}) \right)}{A(\pi^t(\ell(s)|s), \tilde{\pi}^t(\ell(s)|s))}$$

$$\leq \gamma \sum_{t=1}^{T} \underbrace{\left| \bar{Q}(s, \pi^{t-1}, c^{t-1}) + \bar{Q}(s, \pi^t, c^{t-1}) \right|}_{\leq 4} \left| \bar{Q}(s, \pi^{t-1}, c^{t-1}) - \bar{Q}(s, \pi^t, c^{t-1}) \right| \underbrace{\left| A^{-1}(\pi^t(\ell(s)|s), \tilde{\pi}^t(\ell(s)|s)) \right|}_{\leq 8}$$

$$\leq 32\gamma \sum_{t=1}^{T} \left| \bar{Q}(s, \pi^{t-1}, c^{t-1}) - \bar{Q}(s, \pi^t, c^{t-1}) \right|$$

$$\overset{\text{Lemma 3.9}}{\leq} 32 \cdot 48\gamma^2 T \log n.$$

Therefore

$$\text{Term II} \leq 4 \cdot 192 \cdot 28\gamma^2 T + 32 \cdot 48\gamma^2 T \log n + 32\gamma.$$

Therefore, neglecting constants, and plugging in the bounds in Equation (16), we obtain

$$\mathcal{R}_{\text{loc}}^T(s) \leq \mathcal{O}\left( \frac{\log T}{\gamma} + \gamma^2 T \log n \right)$$

Therefore by our selection of $\gamma := \mathcal{O}\left( \log^{1/3} T / (T^{1/3} \log^{1/3} n) \right)$ we get

$$\mathcal{R}_{\text{loc}}^T(s) \leq \mathcal{O}\left( (\log(T))^{\frac{2}{3}} (\log n)^{\frac{1}{3}} T^{\frac{1}{3}} \right)$$

$\square$

## B.5  Proof of Therem 2.3

**Theorem 2.3.** Let $\mathcal{D}$ be the $n$-dimensional simplex, $\mathcal{D} = \Delta_n$. There exists an online learning algorithm $\mathcal{A}$ (Algorithm 3) such that for any cost-vector sequence $c_1, \ldots, c_T \in [-1, 1]^n$,

$$\sum_{t=1}^{T} (c^{t-1} + c^t)^\top x^t - \min_{x^* \in \mathcal{D}} \sum_{t=1}^{T} (c^{t-1} + c^t)^\top x^* \leq \mathcal{O}\left( T^{1/3} \cdot \log^{4/3}(nT) \right)$$

where $x^t = \mathcal{A}_t(c^1, \ldots, c^{t-1})$.

*Proof.* By Lemma 3.8 we obtain that $\mathcal{R}^{\text{alt}}(T) \leq H \max_{s \in V} \mathcal{R}^T_{\text{loc}}(s)$

Then, recalling that by construction $H = \log n$ and using the bound on $\mathcal{R}^T_{\text{loc}}(s)$ in Lemma 3.10 gives

$$\mathcal{R}^{\text{alt}}(T) \leq (\log n) \cdot \mathcal{O}\left((\log(T))^{\frac{2}{3}}(\log n)^{\frac{1}{3}}T^{\frac{1}{3}}\right) = \mathcal{O}\left((\log(T))^{\frac{2}{3}}(\log n)^{\frac{4}{3}}T^{\frac{1}{3}}\right)$$

$\square$

# C Omitted Proof of Section 4

In this section we present the omitted proofs of Section 4.

## C.1 Proof of Lemma 4.1

To simplify notation we denote $\hat{c}_t := c^t + c^{t-1}$ for $t \geq 1$ where $c_0 = (0, \ldots, 0)$. Moreover we denote with $\|\cdot\|$ the euclidean norm $\|\cdot\|_2$. Adaptive FTRL (Algorithm 5) admits the following equivalent form.

---
**Algorithm 5** Adaptive FTRL

---

1: **for** round $t = 1, \ldots, T$ **do**

2:    The learner computes $r_{0:t-1} \leftarrow \sqrt{1 + \sum_{s=1}^{t-1}\|\hat{c}_s\|}$

3:    The learner **plays** $w_t \leftarrow \text{argmin}_{\|x\| \leq 1}\left[\sum_{s=1}^{t-1}\hat{c}_t^\top x + \frac{r_{0:t-1}}{2}\|x\|^2\right]$

4:    The adversary **selects** cost $\hat{c}_t$ with $\|\hat{c}_t\|_2 \leq 2$ and the learner **receives** cost $\hat{c}_t^\top \cdot x^t$.

5: **end for**

---

**Lemma C.1** ([5]). *Let $w_1, \ldots, w_T \in \mathcal{B}(0,1)$ the sequence of points produced by Adaptive FTRL given as input the cost-vector sequence $\hat{c}_1, \ldots, \hat{c}_T$ and $x^* := \text{argmin}_{x \in \mathcal{B}(0,1)}\left[\sum_{t=1}^T \hat{c}_t^\top x\right]$. Then for any index $S \in [T]$,*

$$\sum_{t=1}^S \hat{c}_t^\top(w_{S+1} - x^*) + \sum_{t=S+1}^T \hat{c}_t^\top(w_t - x^*) \leq \frac{r_{0:S}}{2}\left(\|x^*\|^2 - \|w_{S+1}\|^2\right)$$

$$+ \sum_{t=S+1}^T \left[\frac{r_t}{2}\left(\|x^*\|^2 - \|w_{t+1}\|^2\right)\right] + \sum_{t=S+1}^T \hat{c}_t^\top(w_t - w_{t+1})$$

*where $r_t = r_{0:t} - r_{0:t-1}$ for $t \geq 1$.*

*Proof.* Let $f_t(x) := \hat{c}_t^\top x + \frac{r_t}{2}\|x\|^2$ where $r_0 = 1$ and $\hat{c}_0 = 0$. Let us also define $f_{0:t}(x) := \sum_{t=0}^t f_t(x)$. Since $\hat{c}_0 = 0$ we get that $f_{0:t}(x) = \sum_{s=1}^t \hat{c}_s^\top x + \frac{r_{0:t}}{2}\|x\|^2$ and thus $w_{t+1} := \text{argmin}_{x \in \mathbb{B}(0,1)} f_{0:t}(x)$. Then,

$$\begin{aligned}
f_{0:T}(x^*) &\geq f_{0:T}(w_{T+1}) \\
&= f_T(w_{T+1}) + f_{0:T-1}(w_{T+1}) \\
&\geq f_T(w_{T+1}) + f_{0:T-1}(w_T) \\
&\geq \sum_{t=S+1}^T f_t(w_{t+1}) + f_{0:S}(w_{S+1})
\end{aligned}$$

As a result we get that,

$$\sum_{t=0}^T \left(\hat{c}_t^\top x^* + \frac{r_t}{2}\|x^*\|^2\right) \geq \sum_{t=S+1}^T \left(\hat{c}_t^\top w_{t+1} + \frac{r_t}{2}\|w^{t+1}\|^2\right) + \sum_{t=0}^S \left(\hat{c}_t^\top w_{S+1} + \frac{r_t}{2}\|w_{S+1}\|^2\right)$$

By rearranging the terms and using the fact that $\hat{c}_0 = 0$ and $r_0 = 1$ we get that,

$$\sum_{t=1}^{S} \hat{c}_t^\top (w_{S+1} - x^*) + \sum_{t=S+1}^{T} \hat{c}_t^\top (w_t - x^*) \le \frac{r_{0:S}}{2} \left( \|x^*\|^2 - \|w_{S+1}\|^2 \right)$$

$$+ \sum_{t=S+1}^{T} \left[ \frac{r_t}{2} \left( \|x^*\|^2 - \|w_{t+1}\|^2 \right) \right] + \sum_{t=S+1}^{T} \hat{c}_t^\top (w_t - w_{t+1})$$

$\square$

**Lemma C.2** ([6])**.** *Let* $w_1, \ldots, w_T \in \mathcal{B}(0,1)$ *the sequence of points produced by Adaptive FTRL given as input the cost-vector sequence* $\hat{c}_1, \ldots, \hat{c}_T$ *and* $x^* := \operatorname{argmin}_{x \in \mathcal{B}(0,1)} \left[ \sum_{t=1}^{T} \hat{c}_t^\top x \right]$. *Then,*

$$\sum_{t=1}^{T} \hat{c}_t^\top w_t - \sum_{t=1}^{T} \hat{c}_t^\top x^* \le 4.5 \sqrt{1 + \sum_{t=1}^{T} \|\hat{c}_t\|^2}$$

*Proof.* Applying Lemma C.1 with $S = 0$ we get that,

$$\sum_{t=1}^{T} \hat{c}_t^\top (w_t - x^*) \quad \le \quad \sum_{t=1}^{T} \frac{r_t}{2} \left( \|x^*\|^2 - \|w_{t+1}\|^2 \right) + \sum_{t=1}^{T} \hat{c}_t^\top (w_t - w_{t+1}) \tag{20}$$

$$\le \quad \frac{r_{0:T}}{2} + \sum_{t=1}^{T} \hat{c}_t^\top (w_t - w_{t+1}) \tag{21}$$

$$\le \quad 0.5 \sqrt{1 + \sum_{t=1}^{T} \|\hat{c}_t\|^2} + \sum_{t=1}^{T} \hat{c}_t^\top (w_t - w_{t+1}) \tag{22}$$

Up next we bound the second term. Let $f_t(x) := \hat{c}_t^\top x + \frac{r_t}{2}$. By Lemma 7 in [31] for $f_1 := f_{0:t-1}$ and $f_2 := f_{0:t}$. Since $f_1$ is 1-strongly convex with respect to the norm $r_{0:t-1} \|x\|^2$ and $f_2 - f_1$ is convex and $2\|c^t\|$-Lipschitz. Then since $w_t := \operatorname{argmin}_{x \in \mathcal{B}(0,1)} f_1(x)$ and $w_{t+1} := \operatorname{argmin}_{x \in \mathcal{B}(0,1)} f_2(x)$, Lemma 7 in [31] implies that

$$\|w_t - w_{t+1}\| \le \frac{2\|\hat{c}_t\|}{r_{0:t-1}} \le \frac{2\|\hat{c}_t\|}{\sqrt{1 + \sum_{s=1}^{t-1} \|\hat{c}_s\|^2}}$$

As a result, we get that

$$\hat{c}_t^\top (w_t - w_{t+1}) \le \|\hat{c}_t\| \|w_t - w_{t+1}\| \le \frac{2\|\hat{c}_t\|^2}{\sqrt{1 + \sum_{s=1}^{t-1} \|\hat{c}_s\|^2}} \le \frac{2\|\hat{c}_t\|^2}{\sqrt{1 + \sum_{s=1}^{t} \|\hat{c}_s\|^2}}$$

Summing from $t = 1$ to $T$, we get that

$$\sum_{t=1}^{T} \hat{c}_t^\top (w_t - w_{t+1}) \le 4 \sqrt{1 + \sum_{t=1}^{T} \|\hat{c}_t\|^2}$$

$\square$

**Lemma C.3.** *Let* $w_1, \ldots, w_T \in \mathcal{B}(0,1)$ *the sequence of points produced by Adaptive FTRL given as input the cost-vector sequence* $\hat{c}_1, \ldots, \hat{c}_T$. *Let any round* $t^* \in [T]$ *such that for all* $t \ge t^* + 1$,

$$\|\sum_{s=1}^{t} \hat{c}_s\| \ge \frac{1}{4} \|\hat{c}_s\|^2 \quad and \quad \sum_{s=1}^{t} \|\hat{c}_s\|^2 \ge 17$$

*Then* $\|w_t\| = 1$ *for all* $t \ge t^* + 1$ *and additionally,*

$$\sum_{t=t^*}^{T-1} \hat{c}_t^\top \cdot (w_t - w_{t+1}) \le \log(1 + T).$$

*Proof.* To simplify notation we denote $\hat{\sigma}_t := \|\hat{c}_t\|^2$ Moreover we denote $\hat{c}_{1:t} = \sum_{s=1}^t \hat{c}_s$ and $\hat{\sigma}_{1:t} = \sum_{s=1}^t \hat{\sigma}_s$. By the definition of $t^* \in [T]$ we know that for all $t \geq t^* + 1$,

$$\frac{\|\hat{c}_{1:t}\|}{\sqrt{1 + \hat{\sigma}_{1:t}}} \geq \frac{\hat{\sigma}_{1:t}}{4\sqrt{1 + \hat{\sigma}_{1:t}}} \geq 1$$

where the last inequality follows by the fact that $\sigma_{1:t} \geq 17$. Since $w_t \in \mathcal{B}(0,1)$ the latter implies that $\|w_t\| = 1$ for all $t \geq t^* + 1$ and thus,

$$w_t = -\frac{\hat{c}_{1:t-1}}{\|\hat{c}_{1:t-1}\|} \quad \text{and} \quad w_{t+1} = -\frac{\hat{c}_{1:t}}{\|\hat{c}_{1:t}\|}$$

$$
\begin{aligned}
\|w_t - w_{t+1}\| &= \|\frac{\hat{c}_{1:t-1}}{\|\hat{c}_{1:t-1}\|} - \frac{\hat{c}_{1:t}}{\|\hat{c}_{1:t}\|}\| \\
&\leq \|\frac{\hat{c}_{1:t-1}}{\|\hat{c}_{1:t-1}\|} - \frac{\hat{c}_{1:t-1}}{\|\hat{c}_{1:t}\|}\| + \|\frac{\hat{c}_{1:t-1}}{\|\hat{c}_{1:t}\|} - \frac{\hat{c}_{1:t}}{\|\hat{c}_{1:t}\|}\| \\
&\leq \|\hat{c}_{1:t-1}\| \cdot \|\frac{1}{\|\hat{c}_{1:t-1}\|} - \frac{1}{\|\hat{c}_{1:t}\|}\| + \frac{\|\hat{c}_t\|}{\|\hat{c}_{1:t}\|} \\
&\leq \frac{\|\hat{c}_{1:t}\| - \|\hat{c}_{1:t-1}\|}{\|\hat{c}_{1:t}\|} + \frac{\|\hat{c}_t\|}{\|\hat{c}_{1:t}\|} \\
&\leq 2\frac{\|\hat{c}_t\|}{\|\hat{c}_{1:t}\|}
\end{aligned}
$$

where the last inequality follows by the triangle inequality, $\|\hat{c}_{1:t}\| \leq \|\hat{c}_{1:t-1}\| + \|\hat{c}_t\|$. As a result,

$$\|w_t - w_{t+1}\| \leq \frac{2\|\hat{c}_t\|}{\|\hat{c}_{1:t}\|} \leq \frac{8\|\hat{c}_t\|}{\hat{\sigma}_{1:t}}$$

where the last inequality follows by the fact that $t \geq t^* + 1$ and thus $\|\hat{c}_{1:t}\| \geq \frac{1}{4}\hat{\sigma}_{1:t}$. Finally we get that,

$$
\begin{aligned}
\sum_{t=t^*+1}^T \hat{c}_t^\top (w_t - w_{t+1}) &\leq \sum_{t=t^*+1}^T \|\hat{c}_t\| \|w_t - w_{t+1}\| \\
&\leq \sum_{t=t^*+1}^T \frac{8\|\hat{c}_t\|^2}{1 + \hat{\sigma}_{1:t}} \\
&\leq \sum_{t=t^*+1}^T \frac{8\hat{\sigma}_t}{1 + \hat{\sigma}_{1:t}} \\
&\leq \log\left(1 + \sum_{t=t^*+1}^T \hat{\sigma}_t\right) \\
&\leq \log(1 + T)
\end{aligned}
$$

$\square$

We conclude the section with the proof of Lemma 4.1. We restate the theorem so as to be consistent with the notation of the section.

**Lemma 4.1.** Let $w_1, \ldots, w_T \in \mathcal{B}(0,1)$ the sequence of points produced by Adaptive FTRL given as input the cost-vector sequence $\hat{c}_1, \ldots, \hat{c}_T$. Let $t_1$ denote the maximum index such that

$$\sum_{t=1}^{t_1} \hat{c}_t^\top w_t \geq -\frac{1}{4}\sum_{t=1}^{t_1} \|\hat{c}_t\|^2.$$

Then the followig holds,

$$\sum_{t=1}^T \hat{c}_t^\top w_t - \min_{x \in \mathbb{B}(0,1)} \sum_{t=1}^T \hat{c}_t^\top x \leq 4\sqrt{1 + \sum_{t=1}^{t_1} \|\hat{c}_t\|^2} + \mathcal{O}(\log T)$$

*Proof.* Let $t_2$ denotes the maximum index such that $\sum_{s=1}^{t} \hat{c}_s^\top w_s \leq -\|\hat{c}_{1:t}\|$ and $t_3$ the maximum index such that $\hat{\sigma}_{1:t} \leq 17$ (as in the proof of Lemma C.3). We consider the following 3 mutually exclusive case,

- $\underline{t_2 \geq \max(t_1, t_3)}$:

  Due to the fact that $t_2 \geq t_1$ we have that for any $t \geq t_2 + 1$,

  $$-\|\hat{c}_{1:t}\| \leq \sum_{s=1}^{t} \hat{c}_s^\top w_s \leq -\frac{1}{4}\hat{\sigma}_{1:t}$$

  where the first inequality follows by the definition of $t_2$ while the second by the definition of $t_1$, $\sum_{s=1}^{t} \hat{c}_s^\top w_s \leq -\frac{1}{4}\hat{\sigma}_{1:t}$ for all $t \geq t_1 + 1$. Since $t_2 \geq t_3$ we additionally get that $\hat{\sigma}_{1:t} \geq 17$ for all $t \geq t_2 + 1$. As a result,

  $$\|\hat{c}_{1:t}\| \geq \frac{1}{4}\hat{\sigma}_{1:t} \quad \text{and} \quad \hat{\sigma}_{1:t} \geq 17 \qquad \text{for all } t \geq t_2 + 1$$

  Meaning that the conditions of Lemma C.3 are satisfied for all $t \geq t_2 + 1$ and thus

  $$\sum_{t=t_2+1}^{T} \hat{c}_t^\top (w_t - w_{t+1}) \leq \log(1+T) \text{ and } \|w_t\| = 1 \quad \text{for all } t \geq t_2 + 1 \qquad (23)$$

  Up next we analyze the regret of Adaptive FTRL,

  $$\sum_{t=1}^{T} \hat{c}_t^\top (w_t - x^*) = \sum_{t=1}^{t_2} \hat{c}_t^\top (w_t - x^*) + \sum_{t=t_2+1}^{T} \hat{c}_t^\top (w_t - x^*) \qquad (24)$$

  $$= \sum_{t=1}^{t_2} \hat{c}_t^\top (w_t - x_{t_2+1}) + \sum_{t=1}^{t_2} \hat{c}_t^\top (w_{t_2+1} - x^*)$$

  $$+ \sum_{t=t_2+1}^{T} \hat{c}_t^\top (w_t - x^*) \qquad (25)$$

  $$\leq -\|\hat{c}_{1:t_2}\| - \sum_{t=1}^{t_2} \hat{c}_t^\top x_{t_2+1} + \sum_{t=1}^{t_2} \hat{c}_t^\top (x_{t_2+1} - x^*)$$

  $$+ \sum_{t=t_2+1}^{T} \hat{c}_t^\top (w_t - x^*) \qquad (26)$$

  $$\leq \sum_{t=1}^{t_2} \hat{c}_t^\top (w_{t_2+1} - x^*) + \sum_{t=t_2+1}^{T} \hat{c}_t^\top (w_t - x^*) \qquad (27)$$

  $$\leq \frac{r_{0:t_2}}{2}(\|x^*\|^2 - \|w_{t_2+1}\|^2)$$

  $$+ \sum_{t=t_2+1}^{T} \frac{r_t}{2}(\|x^*\|^2 - \|w_{t+1}\|^2) + \sum_{t=t_2+1}^{T} \hat{c}_t^\top (w_t - w_{t+1}) \qquad (28)$$

  $$= \frac{r_{0:t_2}}{2}(\|x^*\|^2 - 1) + \sum_{t=t_2+1}^{T} \frac{r_t}{2}(\|x^*\|^2 - 1)$$

  $$+ \sum_{t=t_2+1}^{T} \hat{c}_t^\top (w_t - w_{t+1}) \qquad (29)$$

  $$\leq \sum_{t=t_2+1}^{T} \hat{c}_t^\top (w_t - w_{t+1}) \leq \log(1+T) \qquad (30)$$

  where Inequality (9) follows by the definition of $t_2$ i.e. $\sum_{t=1}^{t_2} \hat{c}_t^\top x^t \leq -\|\hat{c}_{1:t_2}\|$. Inequality (10) follows by the fact that $\sum_{t=1}^{t_2} \hat{c}_t^\top x_{t_2+1} \geq -\hat{c}_{1:t_2}$. Inequality (11) follows by applying Lemma C.1 for $S := t_2$. Equality (12) and Inequality (13) follow by Equation 23.

- $t_1 \geq \max(t_2, t_3)$: By using the exact same arguments as above we can establish that

$$\sum_{t=t_2+1}^{T} \hat{c}_t^\top (x^t - x^{t+1}) \leq \log(1+T) \text{ and } \|x_t\|_2 = 1 \quad \text{for all } t \geq t_1 + 1 \qquad (31)$$

Using the exact same arguments as above we conclude that

$$
\begin{aligned}
\sum_{t=1}^{T} \hat{c}_t^\top (w_t - x^*) &= \sum_{t=1}^{t_1} \hat{c}_t^\top (w_t - x^*) + \sum_{t=t_1+1}^{T} \hat{c}_t^\top (w_t - x^*) \\
&= \sum_{t=1}^{t_1} \hat{c}_t^\top (w_t - w_{t_1+1}) + \sum_{t=1}^{t_1} \hat{c}_t^\top (w_{t_1+1} - x^*) + \sum_{t=t_1+1}^{T} \hat{c}_t^\top (w_t - x^*) \\
&\leq 4.5\sqrt{1 + \hat{\sigma}_{1:t_1}} + + \sum_{t=1}^{t_1} \hat{c}_t^\top (w_{t_1+1} - x^*) + \sum_{t=t_1+1}^{T} \hat{c}_t^\top (w_t - x^*) \\
&\leq 4.5\sqrt{1 + \sigma_{1:t_1}} + \log(1+T)
\end{aligned}
$$

where the first inequality follows by applying Lemma C.2 for $T = t_1$ and the second by repeating Inequalities $(11) - (15)$.

- $t_2 \geq \max(t_1, t_3)$: By the exact same arguments as in the previous case,

$$\sum_{t=1}^{T} \hat{c}_t^\top (w_t - x^*) \leq 4.5\sqrt{1 + \sigma_{1:t_3}} + \log(1+T) \leq 4.5\sqrt{18} + \log(1+T)$$

where the last inequality follows by the fact that $\sigma_{1:t_3} \leq 17$ (definition of $t_3$).

As a result, we have established that in any case,

$$\sum_{t=1}^{T} \hat{c}_t^\top (w_t - x^*) \leq 4.5\sqrt{1 + \sum_{t=1}^{t_1} \|\hat{c}_t\|_2^2} + \log(1+T) + 4.5\sqrt{18}$$

$\square$

## C.2 Proof of Lemma 4.3

To simplify notation we summarize the Step 7 of Algorithm 4 in Algorithm 6.

---

**Algorithm 6** OGD with Shrinking Domain

---

1: $p_1 \leftarrow 0, D_1 \leftarrow [0, 1]$
2: **for** $t = 1 \ldots T$ **do**
3:     The learner **plays** $p_t \in D_t$
4:     The adversary **selects** $z_t$ and $\sigma_t \leq 1$.
5:     The learner updates the interval $D_t \subseteq [0, 1]$ as follows,

$$D_t \leftarrow \left[0, \min\left(1, \frac{\lambda}{\sqrt{1 + \sum_{s=1}^{t} \sigma_s}}\right)\right]$$

    and its actions $p_{t+1} \in [0, 1]$ as follows

$$p_{t+1} \leftarrow [p_t - \eta_t \cdot z_t]_{D_t}$$

6: **end for**

---

**Remark C.4.** We remark that Algorithm 6 corresponds to Step 7 of Algorithm 4 once

$$\lambda := 20, \ z_t := (c^t + c^{t-1})^\top \cdot (w_t + c^{t-1}) \text{ and } \sigma_t := \|c^t + c^{t-1}\|^2$$

**Definition C.5.** A sequence $q_1, \ldots, q_T \in [0, 1]$ is valid in hindsight if and only if there exists a round $t^* \in [T]$ and a $\delta \in [0, 1]$ such that the following hold,

1. $q_t = \delta \cdot \mathrm{I}\,[t \leq t_1]$ ($q_t = \delta$ for all $t \leq t^*$ and $q_t = 0$ for all $t \geq t^* + 1$).

2. At the switching point $t^* \in [T]$,

$$\delta^2 \leq \frac{\lambda^2}{1 + \sum_{t=1}^{t^*} \sigma_t}$$

In Theorem C.6 we present the payoff guarantees of Algorithm 6 with respect to any sequence $q_t$ that is valid in hindsight.

**Theorem C.6** ([5])**.** *Let* $p_1, \ldots, p_T \in [0, 1]$ *a sequence of points produced by Algorithm 6 given as input the sequence* $(z_1, \sigma_1), \ldots, (z_T, \sigma_T)$*. In case* $z_t^2 \leq 4\sigma_t$ *for all rounds* $t \in [T]$ *then for any valid in hindsight sequence* $q_1, \ldots, q_T \in [0, 1]$ *(Definition C.5) the following holds,*

$$\sum_{t=1}^{T} z_t (p_t - q_t) \leq \lambda \left( 1 + 3 \log \left( 1 + \sum_{t=1}^{T} \sigma_t \right) \right)$$

We conclude the section with the proof of Lemma 4.3.

**Lemma 4.3.** Let the sequence of cost-vector $c^1, \ldots, c^T$ given to Algorithm 4 and the produced sequences $x^1, \ldots, x^t \in \Delta_n$ and $p_1, \ldots, p_T \in (0, 1)$. Additionally let $t_1$ denote the maximum time such that

$$\sum_{s=1}^{t} (c^s + c^{s-1})^\top \cdot w_s \geq -\frac{1}{4} \sum_{s=1}^{t} \|c^s + c^{s-1}\|_2^2$$

and consider the sequence $q_t := \mathrm{I}\,[t \leq t_1] \cdot \left( 20 / \sqrt{400 + \sum_{t=1}^{t_1} \|c^t + c^{t-1}\|_2^2} \right)$. Then the following holds,

$$\sum_{t=1}^{T} (c^{t-1} + c^t)^\top (w_t + c^{t-1}) \cdot q_t - \sum_{t=1}^{T} (c^{t-1} + c^t)^\top (w_t + c^{t-1}) \cdot p_t \leq \mathcal{O}(\log T)$$

*Proof.* The sequence $q_t$ is a valid sequence with switching point $t^* := t_1$ and

$$\delta := \frac{20}{\sqrt{400 + \sum_{t=1}^{t_1} \|c^t + c^{t-1}\|_2^2}}$$

Now the sequence $p_t$ produced by Algorithm 4 in Steps 7 and Steps 8 can be viewed as the output of Algorithm 6 with of the input sequence $z_t := (c^t + c^{t-1})^\top \cdot (w_t + c^{t-1})$ and $\sigma_t := \|c^t + c^{t-1}\|^2$. Since

$$\delta^2 \leq \frac{\lambda^2}{1 + \sum_{t=1}^{t^*} \sigma_t}$$

Lemma 4.3 follows by Theorem C.6. $\qquad\square$

