# OpenReview forum: "Alternation makes the adversary weaker in two-player games"
_NeurIPS.cc/2023/Conference — NeurIPS 2023 spotlight_

### Official Review · Reviewer_waxV · 2023-06-20

**Soundness:** 4 excellent
**Presentation:** 4 excellent
**Contribution:** 4 excellent
**Rating:** 7
**Confidence:** 4

**Summary:**

The authors discuss a model of no-regret learning that simulates the use of *alternating* regret updates in a two-player game. In this model, the authors show that the classical lower bound of $\Theta(\sqrt{T})$ regret does not hold, and in fact, it is possible to achieve $\tilde O(T^{1/3})$ regret.

**Strengths:**

I enjoyed reading this paper. It addresses a gap that has been observed empirically many times (the fact that alternating regret updates improves practical performance), and does so in a theoretically very nice way. The paper is also well written and easy to understand. I vote to accept.

**Weaknesses:**

I see no major issues, and I have only a few relatively minor questions; see the next section.

**Questions:**

1. Maybe some typos?
    * Remark 2.5: should that refer to Alg 4 instead of Alg 2? The stated bound for Alg 2 on $\Delta^2$ is $\tilde O(T^{1/3})$.
    * Algorithm 3, line 6 ($\pi^t(s) = \dots$): should there be a factor of $\gamma$ on the first $2q(s, \pi^t, c^{t-1})^\top x$, to match it with Alg 2, Line 3? If not, ignore my Q2 below, as this is no longer an analogue of CFR, but in that case I would appreciate some intuition as to why $\gamma$ is suddenly gone from that term.
1. I think your algorithm for the n-dimensional simplex looks a lot like a version of CFR ([1]; see [2] for a more modern exposition) applied on the binary tree $T$ with Algorithm 2 as the local regret minimizer. This algorithm, then, generalizes immediately to extensive-form games. Does the analysis also generalize? If so, that'd be nice to state.
1. It is quite surprising to me that there is a gap in the power of $T$ between the $\ell_2$ ball and the simplex. Such a gap certainly doesn't exist in usual models of no-regret learning. Is there any intuition for this gap? Do you believe it to be fundamental/do you have a lower bound against achieving $\tilde O(1)$ regret in the simplex case? For example, what happens if you use Alg 4 as your building-block regret minimizer on $\Delta^2$ within Alg 3?


[1] Martin Zinkevich, Michael Johanson, Michael Bowling, Carmelo Piccione. "Regret Minimization in Games with Incomplete Information", NeurIPS 2007.
[2] Gabriele Farina, Christian Kroer, Tuomas Sandholm. "Online Convex Optimization for Sequential Decision Processes and Extensive-Form Games", AAAI 2019

**Limitations:**

Yes

---

> ### Author Rebuttal · Authors · 2023-08-08
>
> Thank you very much for the work and the valuable comments. We will incorporate your comments/suggestions in the revised version of our work.
>
> 1. Thanks for spotting the typos! -Yes Remark 2.5 refers to Algorithm 4 - $\gamma$ is not missing from line 6 of Algorithm 6 (since we have $R(x)/\gamma$). The typo lies in line 3 of Algorithm 2 since again we have $R(x)/\gamma$ and thus $\gamma$ should be removed from $2c_{t-1}^\top x$
>
> 2. Thank you very much for the suggestion. Yes, our algorithm is very similar with the decomposition used in the CFR algorithm and generalizes for EFGs. We are confident that our analysis may carry over with some possible tweaks. We will include this discussion in the camera ready version.
>
> 3. We also agree that having different alternating regrets for different convex sets is not very intuitive (at least it is not the case in standard OLO). O(1) regret may be achievable for general convex sets, however this might require totally novel techniques. Using Algorithm 4 in Algorithm 3 is a very reasonable idea that we have in fact tried. Unfortunately the analysis of Alg. 3 does not carry over once Alg 4 replaces Alg. 2. The reason is that the outputs of Alg 2 are always in distance $\gamma$ at each step which is not the case for Alg 4.

---

> > ### Comment · Reviewer_waxV · 2023-08-12
> >
> > Thank you for the response. I thought before that this was a good paper, and I still do. I keep my score.

---

> > > ### Author Response · Authors · 2023-08-19
> > >
> > > Dear reviewer,
> > >
> > > Thank you for appreciating our work and for the time spent reviewing it.
> > > We will take into account your comments in our next revision.
> > >
> > > Best,
> > > Authors

---

### Official Review · Reviewer_gHiT · 2023-07-05

**Soundness:** 4 excellent
**Presentation:** 4 excellent
**Contribution:** 3 good
**Rating:** 6
**Confidence:** 4

**Summary:**

This paper consider an online linear optimization (OLO) problem motivated by alternating game play in two-player games. Specifically, after the learner chooses $x^t$ and the adversary selects loss vector $c^t$, the learner suffers loss $(c^t + c^{t-1})x^t$ in the alternating OLO instead of $c^t x^t$ in the standard OLO model. The adversary in the alternating OLO model is weaker than that in the standard OLO model and the main results show that $o(\sqrt{T})$ regret is possible for the following two cases: (1) $\mathcal{O}( (\log n)^{4/3} T^{1/3})$ regret for the $n$-dimensional simplex; (2) $O(\log T)$ regret for a $\ell_2$ ball constraints (and when the agent has only 2 actions).

**Strengths:**

1. The proposed alternating OLO model is interesting and has implication for game-play in two-player games.
2. Although privous works show $o(\sqrt{T})$ regret in some restricted cases such as unconstrained domain or when two players use the same algorithm, the main results in this paper hold for the n-dimensional simplex and against an adversary, which is stronger than privous works. The proofs of the main results are non-trivial.
3. The paper is fairly well-written. The authors provided helpful intuition and high-level ideas in the main body, which makes the paper easy to follow.

**Weaknesses:**

1. The $o(\sqrt{T})$ results for the alternating OLO model only hold for n-dimensional simplex and the ball constraints, but not general convex sets.
2. As the authors pointed out, the current analysis is limited to the linear loss setting, and does not directly extend to general convex losses.
3. There is no lower bound results.

However, I would like to remark that the current results are still very interesting and are important first steps towars solving these open questions.

Some minor comments:

1. Line 108: "...OGD with admits..."
2. ALG2-Line 3: check "2\gamma (c^{t-1})..."
3. Line 195: $x^t$ should be $y^t$?
4. Check line 529-530
5. Check the last inequality on page 22
6. Missing squares in the inequality on line 609
7. The references of inequalities on line 637-639, 643 are broken.

**Questions:**

The results of the paper holds in the adversarial setting. In the game setting, the two players use the same algorithm and are not necessarily adversary to each other. I would like to ask whether better regret guarantees can be achieved in the game setting when the two players use the same algorithm e.g., Algorithm 2 in the paper. Such a result would lead to faster convergence to CCE in two-player games.

---

> ### Author Rebuttal · Authors · 2023-08-08
>
> Thank you very much for your effort and your positive feedback. We will incorporate the minor typos that you pointed to in the revised version of our paper.
>
> *Questions*
>
> *The results of the paper...in two-player games.:* We believe that once both agents use the same algorithm (e.g. Algorithm~2) better bounds to CCE than the current O(T^{-2/3}) can be achieved. This question remained outside the scope of this work since we focused on the adversarial case however it is a very interesting research question.

---

> > ### Comment · Reviewer_gHiT · 2023-08-11
> > **Acknowledgment of Rebuttal**
> >
> > Thank you for your reply. I have no further questions and would like to keep my score.

---

> > > ### Author Response · Authors · 2023-08-19
> > >
> > > Dear reviewer,
> > >
> > > Thanks a lot for your review, for appreciating our work and for acknowledging our rebuttal. We will fix the typos in our next revision.
> > >
> > > Best,
> > > Authors

---

### Official Review · Reviewer_5x6H · 2023-07-06

**Soundness:** 2 fair
**Presentation:** 2 fair
**Contribution:** 3 good
**Rating:** 5
**Confidence:** 3

**Summary:**

The paper considers online linear optimization where losses are given by $x_t^{\top} (c_t + c_{t-1})$, where $c_t$ is the adversary’s choice at each round. The authors show that the standard $\Omega(\sqrt{T})$ lower bound no long holds, and give a $O(T^{1/3})$ regret algorithm over the simplex as well as a $O(\rho \log T)$ regret algorithm over a radius-$\rho$ ball.

**Strengths:**

The paper introduces a new setting for online learning with constrained adversaries, and gives two algorithms with new, non-trivial regret bounds.
The paper is somewhat easy to follow, albeit with some space allocation and presentation issues, and results appear largely correct (although the second algorithm is more difficult to evaluate due to reliance on prior work).

**Weaknesses:**

On the whole, this strikes me as a paper whose core ideas have the potential to be quite compelling if sufficiently extended and contextualized, but which leaves many relevant questions underdiscussed and is not yet ready for publication.

To start, the connection between the costs $x_t^{\top} (c_t + c_{t-1})$ and “alternating gameplay“ could be better motivated; in such turn-based games, why should we think of players as experiencing costs in each round regardless of whose turn it is? Why we should consider this as our cost model for such a game rather than those enabled by e.g. extensive-form games or Stackelberg games? Both of the latter are well-studied from a regret minimization perspective and should be discussed in related work. The formulation given seems more natural as perhaps an example of a more general framework (e.g. when the adversary’s movements are bounded each round) rather than a primary focus in its own right without further contextualization within the broader game theory literature. For example, Remark 1.2 regarding convergence to NE/CCE should probably be omitted unless there is further discussion as to why these solution concepts are appropriate targets for the “alternating” setup (vs. something like Stackelberg equilibria).

Theorem 2.2 is well-known and does not need to be proved in the body; you could maybe state it as a proposition and discuss more informally, but in general, it is not particularly surprising that a specific construction for one setting fails in a more restricted setting, particularly when there are many other known settings where faster rates are obtainable against constrained adversaries (e.g. strongly convex losses).

The alternating setup considered by [WTP22] is a feature of the algorithmic setup and not directly comparable here. It might be more natural to discuss relations to the long line of work on fast convergence rates (e.g. https://arxiv.org/abs/1507.00407). Perhaps there is a deeper connection between fast rates and bounds on the per-round movement of the adversary, which appears in these works as well.

Further, the results are only given for special cases (simplex, ball), and no lower bounds are shown. It seems initially a bit strange that the rates for the simplex and the ball differ so dramatically. Why are the algorithm designs for each so different? The intuition behind the algorithms should be highlighted further, possibly in exchange for moving some proof details to the appendix. Is O(T^1/3) optimal for the simplex? Also, it is difficult to sanity-check the $O(\log T)$ result over the ball for correctness; this result indeed seems surprising for such flexible adversaries, and the technical results should be explained more clearly in the body in order to highlight the primary difficulties and innovations.

Throughout, there are also a number of typos and broken sentences (ex. the statement of “Q2”).


**Questions:**

- What are the barriers to proving fast-rate results for general convex bodies (e.g. in terms of ratio of inner/outer ball)?
- Why is there such a gap between the ball and the simplex?
- If log(T) regret is possible over the ball, it seems natural to expect it’d be possible over any convex body, which would then remove the need for a slower result on the simplex.
- What is the intuition behind the tree design in the simplex algorithm?
- Are there notable technical connections to fast-rates results for optimistic learning dynamics?

**Limitations:**

The connection to concept of “alternation” in games isn’t particularly convincing, and seems like it should be generalizable to broader settings (e.g. movement-bounded adversaries).

Why only the simplex/ball and not convex sets?

Throughout, there are a number of key omissions of important related work, as well as too much emphasis on restating well-known prior results.

---

> ### Author Rebuttal · Authors · 2023-08-08
>
> Thank you very much for your work and valuable comments. Up next we answer the main points raised in the weakness part. We hope that the following discussion (that we will incorporate in our revision) will help reconsidering your suggestion.
>
> 1. *To start...turn it is?:* We remark that alternating game play at which agents update their strategies at a round robin scheme but all experience cost at each round, is a very natural dynamic in game theory and that is aligned with previous works [5-10] (refs in Rev EHgn). For example, best-response dynamics always allow only a single player to update their strategy. Nevertheless, in [5] the performance of no-regret as well as best-response dynamics is considered and in both cases the social cost on each day is the sum of all agents’ cost (not just the ones who moved) showcasing that all agents experience costs every day. Furthermore, intuitively, consider the case of two competing companies which alternatingly adjust to each other's strategies. Clearly, the update of the first company’s policy affects the second’s profits which in turn triggers a further update from the second company and so on.
> 2. *Why we should...related work.:*
> Stackelberg games inherently put one of the two players at a disadvantage. In contrast, in the context of the two company competition alternating game-play captures the competition between two symmetric companies. Alternating game-play can be extended to extensive form games - two poker players alternatingly adjust to each other's strategies over time (a strategy in EFGs is a collection of prob. distr. over the information sets). Extending our results to EFGs is a very interesting question that however remains outside the scope of this work. In the revised version of the paper we will thoroughly discuss the previous works in the above settings.
> 3. *The formulation...“alternating” setup:*  Your suggested framework where the adversary’s movements are bounded within each round *is not* a generalization of our framework. Critically, when our the opponent updates their behavior (i.e. every other time-step) the updates can be *arbitrary*. More precisely, in the Alternating OLO, $||(c_t + c_{t-1}) - (c_{t-1} + c_{t-2})|| = ||c_t - c_{t-2}||$ which can be $\Theta(1)$. We finally remark that NE/CCE are seminal equilibrium concepts in normal-form games regardless of the game-play (simultaneous or alternating).
> 4. *Theorem 2.2...adversaries:* We will rephrase to proposition.
> 5. *The alternating...as well.:* In the current write-up we have briefly discussed the long line of research of learning in games (including the suggested paper) - line 129. However we will include further discussion to better illustrate the differences.
> The major difference with our work consists in the fact that in all the mentioned works the improved regret-guarantees hold *only if all* agents use the exact same algorithm (an optimistic method) which further implies small updates from the opponent. Our improved regret guarantees in the context of alternating game play hold no matter the choices of the adversary (that can be arbitrary away from their previous play). Totally distinct techniques are thus needed for our improved regret bounds.
> 6. *Further...innovations.:* Extending our results to general convex sets is an interesting research direction that remains outside the scope of this work. We are not aware of any lower bound on the alternating regret.
> The unit ball case admits a “simpler” structure compared to simplex permitting a more fine-grained analysis. At an intuitive level the cornerstone difference is that best response in unit ball with respect to $c_{t-1}$ is simply $-c
> _{t-1}/||c_{t-1}||$ while in simplex there might be a whole continuum of optimal points. In the revised version of our work we will discuss in detail the intuition behind our algorithmic design.
>
> *Questions*
>
> 1. *Q1*: (As in Reviewer EHgn)
> 2. *Q2*: As mentioned above, the main reason for this discrepancy comes from the fact that a linear function always admits a unique minimizer in unit ball while it may admit a continuum of optimal points in the case of simplex. This “uniqueness property” of the unit ball permits an interpolation between the output of Adaptive FTRL and the greedy best-response (Step 5 of Alg 4). We remark that extending Alg 4 to simplex is very challenging since there might be not a unique best response with respect to $c_{t-1}$ while all the intermediate technical claims of the analysis of Alg 4 crucially use the simple structure of the unit ball (e.g. projecting a vector $c$ in unit call is just $c/||c||$).\
> That being said, we believe that the provided $O(T^{1/3})$ bound on the alternating regret  for the case of simplex could be improved to $O(\log T)$ or even $O(1)$. However the latter requires new ideas and techniques and remains outside the scope of this work.
> 3. *Q3:*  Yes however it is not clear how an O(log T) alternating regret algorithm for general sets looks like
> 4. *Q4*: The tree decomposition enables us to reduce the problem in the $n$-dimensional simplex to the *easier* 2-dimensional simplex. The reason that in Alg $3$ we use Alg $2$ and not Alg $4$, is that two consecutive outputs of Alg $2$ are always in distance $\gamma$ which is not the case of Alg 4 - see also our response to Reviewer waxV.
> In the revised version of our paper we will discuss in detail the merits of tree decomposition.
> 5. *Q5:* Not really. As mentioned above, the improved regret bounds of  optimistic learning dynamics crucially rely on each agent using a specific algorithm permitting an end-to-end description of the overall system (the key idea is bounded-adversarial movements).  On the opposite front, our improved regret bounds for the alternating setting hold no matter the behavior of the adversary meaning that no modeling on the adversary’s behavior is possible. The latter creates several technical challenges and requires a set of novel ideas and techniques.

---

> > ### Comment · Reviewer_5x6H · 2023-08-14
> >
> > The authors have addressed a number of my criticisms, particularly regarding the setting, and I have increased my score. I still view the the restrictions to the ball/simplex, the mismatched bounds in these cases, and lack of clarity regarding the technical innovations to be weaknesses, and think that the paper is borderline on the whole.

---

> > > ### Author Response · Authors · 2023-08-19
> > >
> > > Dear reviewer,
> > >
> > > Thank you for the time spent reviewing our paper, for answering to our rebuttal, and for increasing the score! We will take your comments into account in our revision.
> > >
> > > Best,
> > > Authors

---

### Official Review · Reviewer_EHgn · 2023-07-29

**Soundness:** 3 good
**Presentation:** 3 good
**Contribution:** 3 good
**Rating:** 7
**Confidence:** 4

**Summary:**

The paper introduces a variant of the usual online learning model with linear losses, whereby the environment is constrained to produce losses of a specific form. Such a form appears naturally when studying alternating no-regret dynamics in games. The authors establish that this restriction weakens the environment, in that the decision maker is now able (at least in certain domains) to attain unconditionally a regret bound of T^1/3, breaking the typical T^1/2 lower bound of regular regret minimization.

**Strengths:**

I believe this is a technically solid paper. While I have some reservations about the relevance of the model per se (that is, assuming adversarial losses of the specific form studied in the paper, which is justified by alternation), I think that taking the model as a given the result are interesting. The paper is also well organized and it is easy to follow the flow with a good understanding of the preliminaries on FTRL and online optimization.

**Weaknesses:**

My only reservation about the paper is about the specific assumption on the losses in general adversarial settings. While the motivation of alternation is clear in two-player games using self-play, I would be curious if the authors could make shed light on what situations their model can model.

The reason why I think the two-player alternation connection is not super satisfying is that in two-player zero-sum games, optimistic (non-alternating) learning dynamics attain constant regret per player. Similarly, in multiplayer general sum games optimistic dynamics guarantee polylogarithmic regret per player when used in self-play. Given that optimism does not come at a computational cost, it is not obvious why, in practice, one would like to focus on alternation but not optimism.

That being said, I have a positive opinion of the technical ideas used in the paper.

Minor comments:
The paper would benefit from a pass of proofreading, as it contains a few typos. A few examples:
- L294: I assume there is a "beyond" missing in the sentence?
- Display after L170: the dots should probably be removed for consistency
- L189: the symbol \mapsto should arguably be a \to in this case, at least according to what I consider standard notation (see also https://math.stackexchange.com/questions/473247/difference-of-mapsto-and-right-arrow).
- Algorithm 3 italicizes keywords such as "select", while Algorithms 1 and 2 bold them.
- L10: I believe it should be "regardless _of_ the strategies", but please double check with a native speaker

**Questions:**

Could you please comment on my reservation above?

Also, I was curious to hear any thoughts you'd like to share about generalizing the algorithm to general convex and compact domains.

**Limitations:**

No negative societal impact concerns

---

> ### Author Rebuttal · Authors · 2023-08-08
>
> Thank you very much for your work and your positive feedback. Thank you also for spotting the typos. Up next, we answer your main points that we will also incorporate in our revision.
>
> 1. *My only reservation...used in the paper* \
> \
> Alternating self-play has proven a valuable optimization tool in several real-life equilibrium computation problems, such as poker as well as generative AI problems, such as GANs [1,2]. From this perspective our work contributes to theoretically understanding the merits of alternation.\
> \
> That being said, the main contribution of our work lies on the game-theoretic perspective. A recent line of works tries to go beyond the o(\sqrt{T}) regret guarantees by either restricting the power of adversary (e.g., small updates [3]) or by using external information of the adversary’s behavior (e.g., hints [4]). \
> \
> Our work reveals that simple alternation between the agents, leads to O(T^{1/3}) (or even O(log T) for 2 actions) regret guarantees no matter the power of the adversary. The latter reveals a sharp phase transition between alternating and simultaneous game play at which  \Omega(T^{1/2}) regret is tight.\
> \
> Finally we would like to remark that various aspects of alternation game-play have been previously considered in the game theory literature [5-10]. For example [9] studies the effect of alternating game-play in cooperative two-player games while it presents several interesting examples at which alternation naturally arises in the games related to animal and social behavior.
>
> $$ $$
>
> 2. *Also, I was curious to...and compact domains.* \
> \
> We believe that Algorithm 2 can be extended to general convex sets - by considering a log-barrier function capturing the convex set of interest. The main challenge in establishing the O(T^{1/3}) alternating regret guarantee for this algorithm is generalizing Lemma 3.3, which quantifies the differences between the output of Algorithm 2 and the output of FTRL, to general convex sets.
>
> $$ $$
>
> 1] Solving Large Imperfect Information Games Using CFR+, Tamelin et al. 2014
>
> [2] Superhuman AI for heads-up no-limit poker: Libratus beats top professionals, Brown et al. 2018
>
> [3] Online Learning: Stochastic, Constrained, and Smoothed Adversaries,  Rakhlin et al. 2013
>
> [4] Online learning with imperfect hints, Bhaskara et al. 2021
>
> [5] Intrinsic robustness of the price of anarchy, Roughgarden et al. 2015
>
> [6] Convergence to approximate nash equilibria in congestion games, Chien et al. 2007
>
> [7] Finite regret and cycles with fixed 309 step-size via alternating gradient descent-ascent, Bailey et al. 2020
>
> [8] Alternating mirror descent for constrained min-max games, Wibisono et al. 2022
>
> [9] Cooperation in alternating interactions with memory constraints, Park et al. 2022
>
> [10] Near-optimal local convergence of alternating gradient descent-ascent for minimax optimization, Zhang et al. 2022

---

> > ### Comment · Reviewer_EHgn · 2023-08-18
> > **Thanks**
> >
> > Thanks for your answers! My opinion of the work was and remains positive.
> >
> > Upon re-reading some parts of the paper I found an additional typo:
> > - L112: where -> here

---

> > > ### Author Response · Authors · 2023-08-19
> > >
> > > Dear reviewer,
> > >
> > > Thanks a lot for all the work on our submission, for your positive evaluation and for all the suggestions during the discussion.
> > > We will take them into account in our revision.
> > >
> > > Best,
> > > Authors

---

### Decision · Program_Chairs · 2023-09-21

**Decision:**

Accept (spotlight)

**Comment:**

This paper proposes a very interesting input model. While the model is not well-motivated in the traditional online learning sense, it is nicely motivated by the practical success of alternation in solving large-scale games and other saddle-point problems. I was somewhat surprised that a result like this holds, actually, since the related work on alternation with mirror descent using a Legendre regularizer explicitly uses the primal-dual dynamics, whereas the present paper gives a worst-case input guarantee under a model that seems to capture inputs generated by alternating no-regret algorithms.